# Proteome-wide analysis of a malaria vaccine study reveals personalized humoral immune profiles in Tanzanian adults

Flavia Camponovo[1,2], Joseph J Campo[3], Timothy Q Le[3], Amit Oberai[3], Christopher Hung[3], Jozelyn V Pablo[3], Andy A Teng[3], Xiaowu Liang[3], B Kim Lee Sim[4], Said Jongo[5], Salim Abdulla[5], Marcel Tanner[1,2], Stephen L Hoffman[4], Claudia Daubenberger[1,2], Melissa A Penny[1,2]*

[1]Swiss Tropical and Public Health Institute, Basel, Switzerland; [2]University of Basel, Basel, Switzerland; [3]Antigen Discovery Inc, Irvine, United States; [4]Sanaria, Rockville, United States; [5]Ifakara Health Institute, Dar es Salaam, United Republic of Tanzania

**Abstract** Tanzanian adult male volunteers were immunized by direct venous inoculation with radiation-attenuated, aseptic, purified, cryopreserved *Plasmodium falciparum* (Pf) sporozoites (PfSPZ Vaccine) and protective efficacy assessed by homologous controlled human malaria infection (CHMI). Serum immunoglobulin G (IgG) responses were analyzed longitudinally using a Pf protein microarray covering 91% of the proteome, providing first insights into naturally acquired and PfSPZ Vaccine-induced whole parasite antibody profiles in malaria pre-exposed Africans. Immunoreactivity was identified against 2239 functionally diverse Pf proteins, showing a wide breadth of humoral response. Antibody-based immune 'fingerprints' in these individuals indicated a strong person-specific immune response at baseline, with little changes in the overall humoral immunoreactivity pattern measured after immunization. The moderate increase in immunogenicity following immunization and the extensive and variable breadth of humoral immune response observed in the volunteers at baseline suggest that pre-exposure reduces vaccine-induced antigen reactivity in unanticipated ways.

**\*For correspondence:**
melissa.penny@unibas.ch

## Introduction

Malaria control and elimination remain a significant public health challenge, and an effective malaria vaccine targeting *Plasmodium falciparum* (Pf) would be an important tool to accelerate burden reduction, curb the spread of drug resistant strains and facilitate focal Pf malaria elimination (*Greenwood, 2008*). While the most advanced vaccine, RTS,S/AS01A, an adjuvanted subunit vaccine for pediatric indications based on the major Pf sporozoite surface protein (*Alonso, 2006*), is currently being assessed in a large pilot study for safety, impact, and feasibility of routine immunization of children (*World Health Organization, 2018*), other vaccines are in development or currently tested. The Malaria Vaccine Roadmap provides guidance for next-generation vaccine development targeting all age groups with improved efficacy and extended duration of protection (*malERA Refresh Consultative Panel on Tools for Malaria Elimination, 2017*). However, although extensive work has been undertaken to understand potential immune mechanisms of vaccine-induced protection against malaria infection and disease, much remains unknown.

An alternative to the subunit vaccine approach is immunization with attenuated whole Pf sporozoite vaccines (*Richie et al., 2015*). Malaria sporozoites have been studied in the context of inducing sterile immunity first in mouse models (*Nussenzweig et al., 1967*), and later in humans via Pf-

infected mosquitoes (*Clyde et al., 1973*; *Rieckmann et al., 1974*; *Hoffman et al., 2002*). Whole sporozoite malaria vaccine development is currently based on aseptic, purified, metabolically active Pf sporozoites (NF54 strain), either radiation-attenuated (PfSPZ Vaccine), chemo-attenuated through concurrent antimalarial administration (PfSPZ-CVac), or genetically modified (PfSPZ-GA2) (*Richie et al., 2015*). Clinical trials have evaluated the efficacy of PfSPZ Vaccine (e.g. *Epstein et al., 2011*; *Seder et al., 2013*; *Ishizuka et al., 2016*; *Epstein et al., 2017*; *Lyke et al., 2017*) resulting in evidence that intravenous injections provide higher protection than intradermal applications against homologous and heterologous CHMI. Specifically, all US adults (n = 6) volunteers were protected against homologous CHMI in a 5-dose schedule (*Seder et al., 2013*), of whom 5 of 6 remained protected following CHMI at 59 weeks after last immunization (*Ishizuka et al., 2016*). Protection following heterologous CHMI (7G8 strain) has been achieved for up to 33 weeks after last vaccine dose (*Epstein et al., 2017*; *Lyke et al., 2017*). These results enabled the PfSPZ Vaccine to receive an FDA Fast Track designation (*Sanaria Inc, 2016*).

To evaluate the protective efficacy against CHMI of PfSPZ Vaccine in malaria pre-exposed volunteers, PfSPZ Vaccine was evaluated in a dose-escalation study in a cohort of adult, male Tanzanian volunteers (acronym: BSPZV1) (*Jongo et al., 2018*). In this study, vaccine-induced protection against homologous CHMI was assessed for the first time in Sub-Saharan Africa by direct venous inoculation (DVI) of 3200 fully infectious sporozoites (PfSPZ Challenge) three and/or 24 weeks after last immunization. 1/18 volunteers who received 5 doses of $1.35 \times 10^5$ PfSPZ (group 2,) and 4/20 volunteers who received 5 doses of $2.7 \times 10^5$ PfSPZ (group 3) were sterilely protected 3 weeks after last vaccine dose. The four individuals from the higher dose group protected in the first CHMI remained protected at a second CHMI conducted 24 weeks after final PfSPZ immunization (*Jongo et al., 2018*).

Our understanding of naturally or vaccine induced cellular immune dynamics has considerably improved over the years, however much remains unknown. The immune mechanisms conferring protection associated with PfSPZ Vaccine have been studied in mouse and non-human primate models, and in human biological specimens (*Lefebvre and Harty, 2020*). In mice, the role of liver resident CD8+, IFNγ producing T cells, and γδT Cells have been considered paramount for conferring protection (*Lefebvre and Harty, 2020*), but their role in humans remain more difficult to assess. Pf-specific T cells found in peripheral blood might not represent the more abundant and stable tissue resident effector T cells (*Ishizuka et al., 2016*), and thus might not represent a good proxy for cellular immunity in the liver. CD4 and CD8 T cells producing IFNg, interleukin 2 (IL2) and tumor necrosis factor alpha (TNFa) were detected in peripheral blood after immunization of malaria naïve volunteers, however, their levels rapidly declined over time and were not correlated with protection (*Ishizuka et al., 2016*; *Lyke et al., 2017*). No increase in CD8 T cell responses was observed following immunization of Tanzanian volunteers, and CD4 levels increased to lower magnitudes compared to malaria naïve individuals, with no association with protection (*Jongo et al., 2019*).

There has been significant work to assess the quality and quantity of antibody responses following PfSPZ Vaccine application. IgG specific to the Pf circumsporozoite protein (CSP) measured by ELISA or detected against PfSPZ by automated immunofluorescence assay (aIFA) correlated with protection in malaria naïve volunteers at 3 week and 21–25 week post-vaccination homologous CHMI in one study (*Ishizuka et al., 2016*). In contrast, the PfCSP ELISA and PfSPZ aIFA trend was not significant when the sporozoite dose was increased to $9 \times 10^5$ PfSPZ administered three times (*Lyke et al., 2017*), and these assays, along with an inhibition of sporozoite invasion (ISI) assay, did not correlate with protection in follow up studies (*Epstein et al., 2017*).

A functional role of antibodies in protection after PfSPZ Vaccination was proposed after passive transfer of the IgG fraction of immune sera from protected volunteers into humanized FRG-huHep mice that led to an 88% and 65% reduction in parasite liver burden with immune sera collected 2–3 weeks and 59 weeks after last PfSPZ immunization, respectively (*Ishizuka et al., 2016*). Further evidence of functional activity of humoral immunity was shown with anti-PfCSP monoclonal antibodies (mAbs) isolated from Tanzanian volunteers (*Tan et al., 2018*) and from U.S. vaccinees (*Kisalu et al., 2018*). Additionally, IgM specific to PfCSP was detected in Tanzanian pre-exposed adult males after immunization with PfSPZ Vaccine, and these IgM inhibited liver cell sporozoite invasion in vitro and fixed complement on whole Pf sporozoite (*Zenklusen et al., 2018*). Antigen-specific humoral immune responses have been studied in clinical trials by ELISA for a selection of antigens, such as PfCSP, PfEXP1, PfEBA-175, PfLSA1, PfMSP1 and PfMSP3 (see *Epstein et al., 2017* for summary of

findings). However, the number of antigens that can be assessed in parallel with ELISA is limited and thus constrained to pre-selected of antigens.

In contrast, malaria protein microarrays enable a less biased approach to assess humoral immunity against a large proportion of Pf encoded proteins (*Liang and Felgner, 2015*). This technology has assessed potential boosting of natural immunity in RTS,S vaccinated individuals (*Campo et al., 2015*), and was used to better understand the natural history of malaria infection in the field (*Boudová et al., 2017*). Microarrays have also identified immune-reactive antigens associated with malaria exposure for immune-epidemiological studies, or to examine potential correlates of protection in children and adults from endemic areas (*Dent et al., 2015*). Serum samples from malaria-naïve individuals infected by radiation-attenuated sporozoites via mosquito bites have been probed with microarrays covering 23% of all Pf proteins, providing insight into humoral immunity induced by both whole sporozoite vaccination and its association with sterile protection following CHMI (*Trieu et al., 2011*). More recently, a whole proteome microarray produced by Antigen Discovery, Inc (ADI) that includes 91% of all predicted Pf 3D7 strain proteins was used to study humoral immunity after PfSPZ-CVac vaccination in European malaria-naive volunteers (*Mordmüller et al., 2017*). 22 proteins were identified, which were recognized by more than 50% of the volunteers in the highest dose group, all of whom were protected against homologous CHMI (*Mordmüller et al., 2017*).

The main *in-vitro*, animal, and human studies described above suggest that both cellular immune response and antibody mediated immune response plays a role in inducing protection. However, a complete understanding of the mechanisms of vaccine-induced protection against malaria infection, and its interplay with pre-built natural immune response in exposed populations, remains unknown. For a comprehensive and unbiased description of Pf-specific humoral immune responses in malaria pre-exposed volunteers, we analyzed serum samples using the Pf protein microarray featuring 7455 full-length or fragmented proteins of the Pf proteome (3D7) (*Mordmüller et al., 2017*). Serum samples were collected pre-vaccination and 14 days past last vaccination to understand the PfSPZ Vaccine-induced IgG profiles in Tanzanian male adults participating in the BSPZV1 study and potential correlations between humoral immune responses and PfSPZ vaccine induced protection against homologous CHMI (*Jongo et al., 2018*). We show for the first time that our study population displayed highly personalized immune profiles based on a broad range of antigens recognized before vaccination. Surprisingly, this humoral immune pattern remained largely unchanged following the whole organism based PfSPZ immunization, leading to the hypothesis of natural imprinting of humoral immune responses.

## Results

### Study volunteers and serum sampling

In total, 92 serum samples from 46 volunteers enrolled in the BSPZV1 study (all volunteers from group 2 and group 3 in the clinical trial [*Jongo et al., 2018*]) were probed on Pf whole proteome microarrays, including samples collected at baseline (before vaccination) and 2 weeks after last immunization (*Figure 1*). Eight non-vaccinated placebo controls, 18 volunteers who were immunized with the lower PfSPZ Vaccine dose (group 2) and 20 volunteers who received the higher PfSPZ Vaccine dose (group 3) were included (*Figure 1*; *Jongo et al., 2018*). All volunteers included in the study had no parasitemia at the start of the study (measured by malaria thick blood smears (TBS)) and no parasitemia before CHMI (measured by TBS and the more sensitive qPCR) (*Jongo et al., 2018*). Additional exclusion criteria included history of malaria in the previous 5 years or antibodies to PfEXP1 by ELISA above a threshold level (*Jongo et al., 2018*) associated with recent infection by CHMI (*Shekalaghe et al., 2014*).

### Tanzanian male adults recognize a high diversity of pf proteins

Across the 7455 Pf full length or fragmented proteins, 2804 probes corresponding to 2239 Pf proteins were considered as reactive antigens in the 92 samples tested for having a seropositive response (normalized signal intensity $\geq$1) in at least 10% of volunteers at either or both time points.

First, we examined the antibody profiles for each volunteer individually and the results of the paired samples are presented in the heatmap (*Figure 2a*). Pattern of antigens recognized across all peptides was largely unchanged before and after immunization. This was further confirmed by

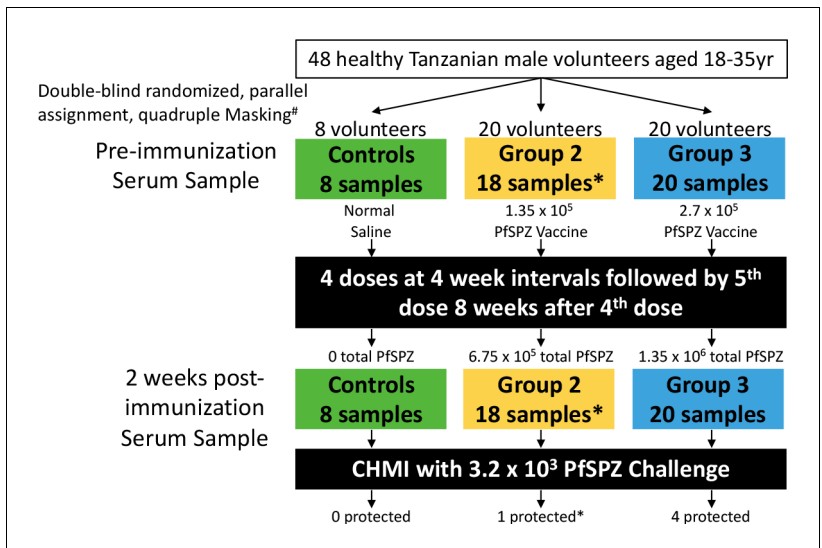

**Figure 1.** Sampling and volunteer information for proteome microarray studies. Three arms of a randomized, double-blind Phase 1 trial of PfSPZ Vaccine were selected for antibody profiling on Pf whole proteome microarrays: normal saline controls, a lower dose (group 2, $1.35 \times 10^5$ PfSPZ Vaccine/dose) and a higher dose (group 3, $2.7 \times 10^5$ PfSPZ Vaccine/dose). Serum samples were collected before immunization and 2 weeks after the final immunization. Information on the protection status of the volunteers after a 3 week post-immunization CHMI is provided. #Masking included participant, care provider, investigator and outcome assessor. *Samples were unavailable for protein array screening from two group 2 volunteers, one did not receive the 5th immunization dose and one left the country before CHMI (*Jongo et al., 2019*). All volunteers in the clinical trial who received 5 doses of immunization and who underwent CHMI 3 weeks after last immunization dose were included in the current analysis.

applying a dimensionality reduction algorithm (t-Distributed Stochastic Neighbor Embedding (t-SNE)) to represent the 2804 probes recognized in two dimensions and including time points assessed (*Figure 2b*). The t-SNE algorithm estimates the probability distribution of neighbors around each point, that is, it models the set of points which are closest to each point. A distinct clustering of observations for 43 out of the 46 volunteers was evident, with each pair of sample's nearest neighbor in the first two dimensions as the corresponding sampling time point for that volunteer (*Figure 2b*). Volunteer samples did not cluster according to treatment allocation with no difference identified by t-SNE between controls and vaccinees, but rather volunteers preserved their immunoreactive 'fingerprint' measured in their samples taken at baseline and two weeks following last PfSPZ Vaccination. It was not obvious why these three subjects did not maintain longitudinally consistent antibody profiles and we cannot rule out technical or sampling issues, or if they are due to unidentified biological factors associated with vaccination or temporal immune status. However, our results show these three individuals had slightly higher than average baseline antibody breadth and steeper decrease of breadth after immunization (*Figure 3c*).

To further investigate the breadth of humoral immune response against the 2804 peptides, the total number of peptides regarded as sero-reactive per sample were analyzed (*Figure 3*). Antigen recognition varied widely among individuals, with breadth of humoral immune response ranging from 187 to 2360 reactive antigens across all samples before immunization, and from 217 to 1535 and 187 to 1965 reactive antigens per volunteer in group 2 and group 3, respectively, after PfSPZ Vaccination (*Figure 3a–b*). Median antibody breadth from the PfSPZ-immunized volunteers across both immunization groups was 720 and 669 reactive peptide features recognized before and after immunization, respectively.

The effect of vaccination on breadth was analyzed by comparing breadth between the control and the PfSPZ Vaccine-immunized groups, and by quantifying the change in breadth after immunization compared to baseline levels. No significant differences in antibody breadth between the control and group 2 or group 3, nor between group 2 and group 3 after vaccination were detected (results of the negative binomial regression summarized in Table supplement 1). During the period before

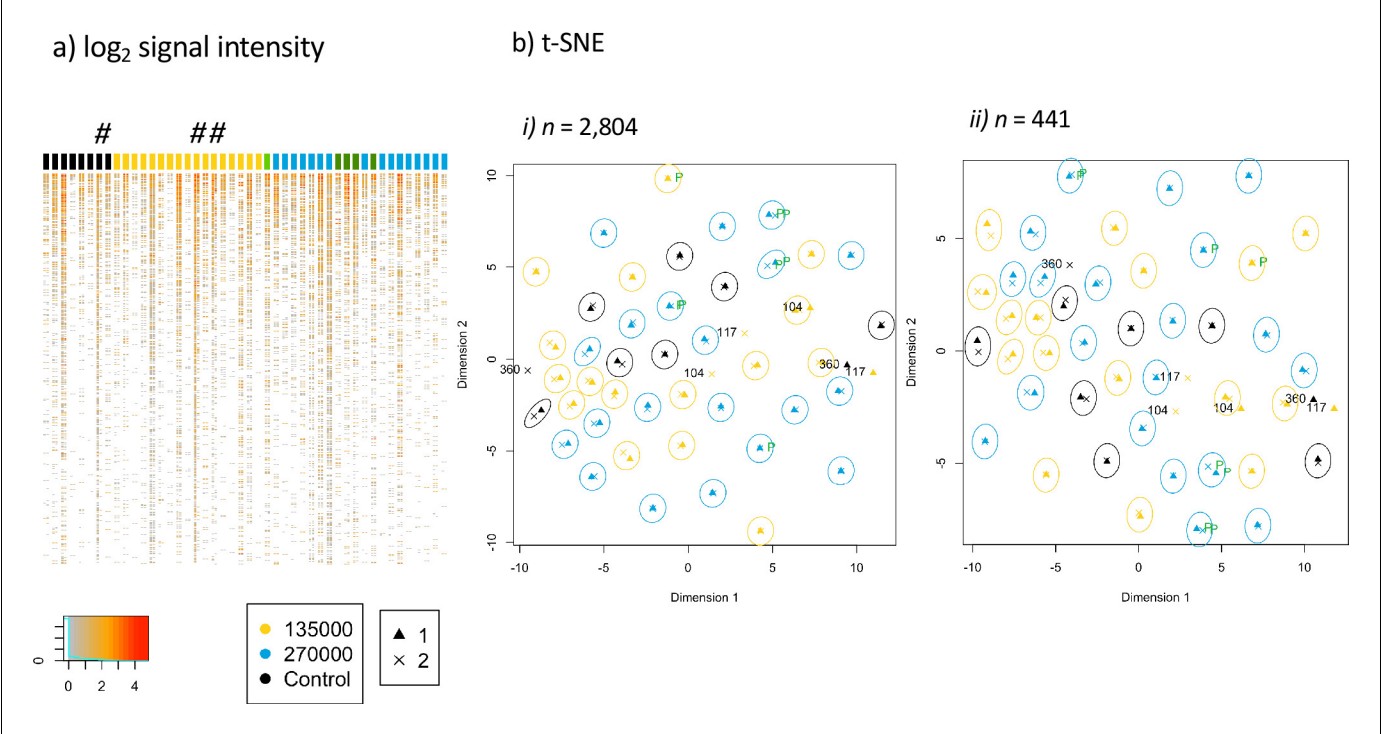

**Figure 2.** Antibody immune profile of Tanzanian healthy male adult volunteers is personalized. The heatmap of the normalized signal intensities of each sample per subject is shown in (a), with the signal intensity of each protein or fragment (rows) displayed for the samples before and after immunization of each volunteer (columns). The colored column headers represent volunteers ordered according to treatment allocation. 8 subjects of the control group (black), CHMI unprotected subjects of group 2 (n = 17, yellow), unprotected subjects of group 3 (n = 16, blue), CHMI protected subjects (n = 5, green). The first and second columns for each subject display the results obtained from baseline and after immunization samples, respectively. The # indicates volunteers BSPZV1-360, BSPZV1-104 and BSPZV1-117. (b) A t-SNE projected dimensionality reduction of normalized signal intensities across the microarray spots measured at baseline (triangles) and after PfSPZ vaccination (crosses) is shown. In (b-i) data are shown for the total 2804 reactive spots and in (b-ii) for the subset of 441 reactive proteins fragments predicted to be expressed at the sporozoite stage (*Florens et al., 2002*), with the signals obtained for each subject at the two bleeding time points grouped in circles. For 3 out of 46 subjects, namely BSPZV1-360, BSPZV1-104 and BSPZV1-117, the signals do not cluster in this t-SNE analysis.

The online version of this article includes the following source data for figure 2:

**Source data 1.** Data frame of the normalized signal intensities of the protein microarray.

and after vaccination, antibody breadth declined in many individuals in the control and immunized groups (note that samples were balanced for group and time point factors across technical microarray factors using a block randomization design, see Materials and methods), the relative differences in the medians between the two time points were of −4.8%, −13.6% and −4.7% seropositive signals in the control, group 2 and group 3, respectively (effect of vaccination on breadth tested with the inverted beta-binomial test for paired count data, resulting in an estimated fold change of −1.23,−1.16 and 1.03, and a p-value of 0.24, 0.03 and 0.43, respectively) (*Figure 3c*). In controls, 2/8 volunteers (25%) had higher antibody breadth after placebo inoculation compared to baseline (*Figure 3c*), and antibody breadth after PfSPZ Vaccine immunization increased in 6/18 (34%) and 12/20 (60%) individuals in group 2 and group 3, respectively (*Figure 2c*). Note that the volunteers for which samples that did not cluster in the t-SNE outputs appear to have higher than average baseline breadth and steeper decrease of breadth after immunization (*Figure 3c*). Overall, there was no dramatic change in breadths between both time points, which aligns with the immune fingerprint analysis in *Figure 2*. There was a small decreased average breadth in group 2 driven by 2 of the three individuals whose samples did not cluster for immune-fingerprinting.

To assess the biological characteristics of this large number of reactive proteins, we used the DeepLoc method for *in silico* prediction of protein subcellular localization using the 3D7 protein amino acid sequences (*Almagro Armenteros et al., 2017*; *Table 1*). Numerous reactive proteins predicted to be exported (n = 53) or cell membrane associated (n = 208) were identified. The

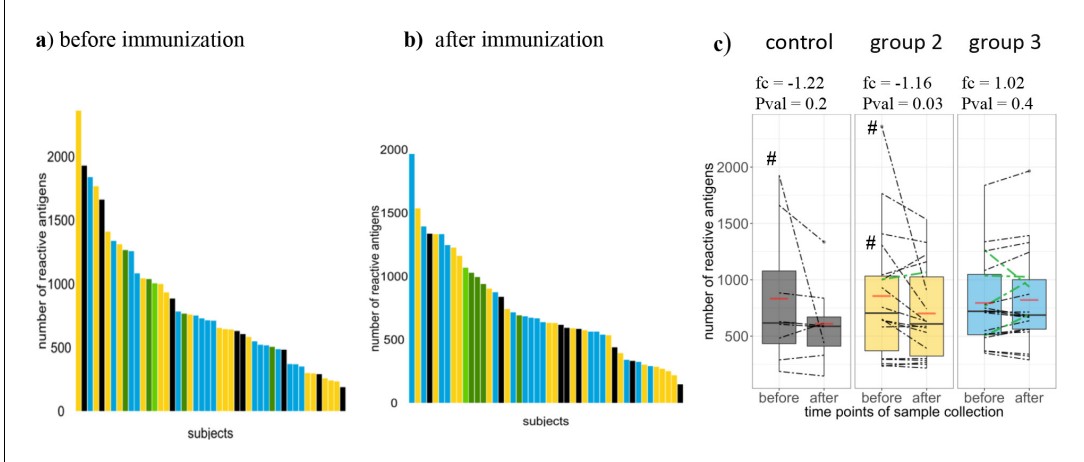

**Figure 3.** Breadth of Pf-specific humoral immunity upon PfSPZ vaccination. Breadth of Pf-specific antibody responses per volunteer (a) before and (b) after PfSPZ vaccination, stratified according to intervention and ordered according to their respective number of seropositive responses from highest to lowest. In (c) boxplots show median, interquartile range (IQR) and 1.5xIQR limits of the antibody breadth grouped by study arm and time point, means for each group are represented by red lines, and an estimated fold change with p-value from the inverted beta-binomial test are indicated for each group. Breadth of each volunteer are indicated by dashed lines. Controls, group 2 ($1.35 \times 10^5$ PfSPZ Vaccine/dose) and group 3 ($2.7 \times 10^5$ PfSPZ Vaccine/dose) volunteers are marked in black, yellow and blue, respectively. Results of the five CHMI protected individuals are highlighted in green (light green in group 2).

The online version of this article includes the following source data for figure 3:

**Source data 1.** Breadth of Pf-specific humoral immunity in each sample.
**Source data 2.** Summary statistics on breadth per group and protection level.

majority of reactive antigens were predicted to be intracellular proteins ($n$ = 1978). However, many of the well-known proteins present in the parasite organelles such as rhoptries and micronemes and proteins exported to the surface of infected erythrocytes such as PfEMP1 variants were predicted to be localized intracellularly, which shows that the currently available prediction algorithms remain limited by complex parasite biology. The full list of reactive antigens and DeepLoc subcellular localization predictions is shown in *Table 1—source data 1*.

## Moderate increase in antigen recognition following immunization with PfSPZ vaccine

To investigate potential vaccine induced increase in antigen immunogenicity, we compared mean reactivity of each antigen 2 weeks after immunization in relation to the baseline responses, grouped according to treatment intervention (*Figure 4a–c*). Reactivity against most antigens decreased after immunization in group 2 and in the placebo group during the ~24 week time interval between the two comparison time points, although not significantly when adjusted for the false discovery rate (FDR) with the Benjamini-Hochberg method (BH) (*Benjamini and Hochberg, 1995*; *Figure 4a,b*), which aligns with the observed slight decrease in breadth of immune response following immunization. Antibodies binding to the Pf circumsporozoite protein (CSP) increased in group 2 and higher in group 3, although not significantly when adjusted for FDR (unadjusted p-value of $4.4 * 10^{-4}$ [adjusted: 0.42] and $2.1 * 10^{-5}$ [adjusted: 0.06] for the t-test in group 2 and group 3, respectively) (*Figure 4c*). The mean negative value for PfCSP immunogenicity at baseline (before immunization, mean intensity = −0.73) suggests that very limited anti-PfCSP responses existed at baseline, with no significant differences between the study groups observed. Notably, the nonsignificant trend for reactivity decline observed in controls in almost all protein fragments of group 2 was not present in group 3.

'Deltas', the fold change in antibody reactivity (averaged across each individual group) for each individual peptide recognized before and after immunization, was higher for PfCSP in group 2 and group 3 when compared to controls, but not statistically significant after adjustment (*Figure 4—figure supplement 2a–b*). In addition, no significant difference was observed by comparing group 2

**Table 1.** Intracellular proteins are the most abundant reactive proteins.

The frequencies of reactive antigens allocated into the different subcellular localization categories (rows) for each group (columns), tested using 2-propotions Z-test and p-values adjusted using the Benjamini-Hochberg method (BH) (*Benjamini and Hochberg, 1995*), are shown (for all reactive proteins with p-values<0.05). Column two indicate the total number of reactive antigens, and columns 3–8 detail the number of significantly differentially reactive proteins localized in each compartment across samples before immunization, after immunization, in the protected group before and after immunization, in the unprotected group before and after immunization, respectively. The first row shows extracellular proteins, the second row is cell membrane associated proteins and the following rows are predicted intracellular proteins split according to subcellular localisation. The percentage of the reactive proteins found in each group compared to all samples (first column) are indicated in parenthesis.

| | Subcellular localization | N reactive proteins | Baseline reactivity | Post-Immz reactivity | Baseline reactivity (protected) | Post-Immz reactivity (protected) | Baseline reactivity (unprotected) | Post-Immz reactivity (unprotected) |
|---|---|---|---|---|---|---|---|---|
| | Extracellular | 53 | 3 (6%) | 3 (6%) | 10 (19%) | 12 (23%) | 3 (6%) | 3 (6%) |
| | Cell membrane | 208 | 11 (5%) | 8 (4%) | 63 (30%) | 70 (34%) | 10 (5%) | 5 (2%) |
| Intracellular (N = 1978) | Cytoplasm | 661 | 14 (2%) | 16 (2%) | 73 (11%) | 79 (12%) | 12 (2%) | 14 (2%) |
| | Endoplasmic reticulum | 429 | 12 (3%) | 11 (3%) | 53 (12%) | 60 (14%) | 11 (3%) | 10 (2%) |
| | Golgi apparatus | 76 | 2 (3%) | 2 (3%) | 13 (17%) | 15 (20%) | 2 (3%) | 2 (3%) |
| | Lysosome/Vacuole | 32 | 1 (3%) | 1 (3%) | 3 (9%) | 2 (6%) | 1 (3%) | 1 (3%) |
| | Mitochondrion | 150 | 3 (2%) | 3 (2%) | 11 (7%) | 14 (9%) | 3 (2%) | 3 (2%) |
| | Nucleus | 624 | 24 (4%) | 20 (3%) | 107 (17%) | 115 (18%) | 17 (3%) | 18 (3%) |
| | Peroxisome | 3 | 0 (0%) | 0 (0%) | 0 (0%) | 0 (0%) | 0 (0%) | 0 (0%) |
| | Plastid | 3 | 2 (67%) | 2 (67%) | 2 (67%) | 2 (67%) | 2 (67%) | 2 (67%) |
| | Total | 2239 | 72 (3%) | 66 (3%) | 335 (15%) | 369 (16%) | 61 (3%) | 58 (3%) |

The online version of this article includes the following source data for Table 1:

Source data 1. The full list of reactive antigens and DeepLoc subcellular localization predictions.

deltas to group 3 deltas *Figure 4—figure supplement 2c*). The nonsignificant trend of higher deltas in group 3 is consistent with the observations of less declining antibodies in the paired analysis.

To identify potential differences in vaccine induced immunogenicity between protected and unprotected individuals, we compared the difference in the mean immunoreactivity (i.e. signal intensity) for each antigen at baseline and 2 weeks after immunization between protected (n = 5) and unprotected (n = 33) volunteers, and the difference in the mean immunogenicity of each antigen between baseline and post immunization time points in the protected group (*Figure 5*). Four proteins were recognized as significantly higher in the protected volunteers after immunization compared to unprotected volunteers (*Figure 5b*). These proteins were the apical membrane antigen 1 (PfAMA1, gene ID PF3D7_1133400) and 3 fragments of the erythrocyte membrane protein 1 (PfEMP1, gene IDs PF3D7_0412900, PF3D7_1240400 and PF3D7_0711700). Interestingly, amino acid sequence alignment of the three identified PfEMP1 protein fragments with the predicted sporozoite encoded variant PF3D7_0809100, recently described to contribute to inhibition of hepatocyte invasion (*Zanghì et al., 2018*), demonstrated long stretches of linear protein sequence conservation (*Figure 5—figure supplement 3*). Nevertheless, the sample size of the protected group is low (n = 5), and considering the low effect size measured by Cohen's distance (*Figure 5—figure supplement 1*), no strong argument for association with protection of these four antigens can be made from this analysis. The mean immunogenicity of PfAMA1 and PfCSP levels across the protected group increased from baseline to after immunization, but not reaching the significance threshold (*Figure 5c*). Notably, a trend of higher reactivity levels to the 3 PfEMP1 fragments in the protected group was observed at baseline, albeit not significantly (*Figure 5a*). No antigen showed an increase in reactivity (delta) significantly higher in the protected group (*Figure 5—figure supplement 2*).

## Breadth of humoral immune response in protected individuals

Analysis above indicates that breadth of humoral immune response was highly variable in samples across all groups and mostly conserved from baseline to post-immunization levels, irrespective of

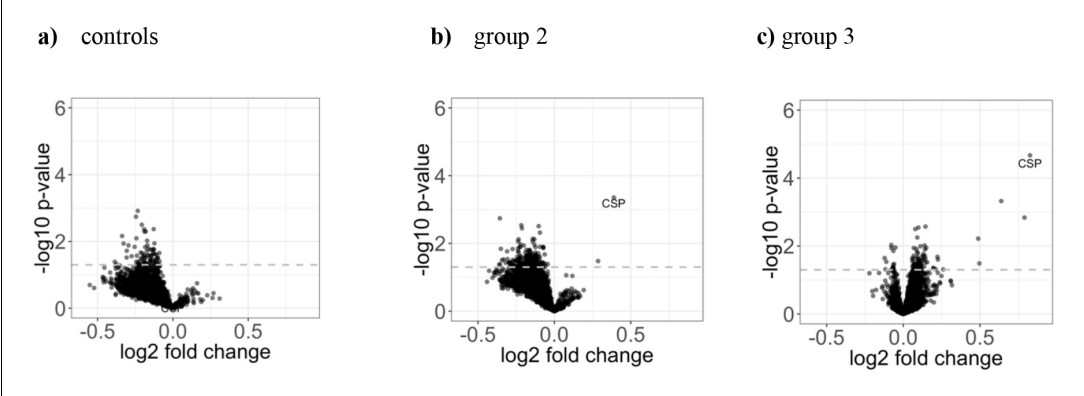

**Figure 4.** Increase in antigen recognition from baseline to after PfSPZ vaccination is moderate. The three volcano plots in the upper row show (**a**) the mean fold change in the control group (n = 8), (**b**) in group 2 (1.35 × 10$^5$ PfSPZ Vaccine/dose) (n = 18), and (**c**) in group 3 (2.7 × 10$^5$ PfSPZ Vaccine/dose) (n = 20). In all groups, the samples collected at baseline and two weeks past last vaccination were compared. The dashed line represents the threshold of statistical significance (p=0.05) not adjusted for the FDR (none of the antigens had a FDR adjusted p-value<0.05). For effect size estimates see *Figure 4—figure supplement 1*.

The online version of this article includes the following source data and figure supplement(s) for figure 4:

**Source data 1.** Source data for plot a.
**Source data 2.** Source data for plot b.
**Source data 3.** Source data for plot c.
**Figure supplement 1.** Effect size for the increase in antigen recognition from baseline to after PfSPZ vaccination.
**Figure supplement 2.** Differential antigen reactivity between control and immunization groups is moderate.
**Figure supplement 2—source data 1.** Source data for plot a.
**Figure supplement 2—source data 2.** Source data for plot b.
**Figure supplement 2—source data 3.** Source data for plot c.
**Figure supplement 3.** Variance and mean of the log$_2$ signal intensities.

protection status (*Figure 3*). We thus further compared breadth of humoral immune response in the protected versus unprotected individuals. Despite a mean breadth in the protected group 12% higher than in the unprotected individuals before immunization (breadth$_{protected}$ = 914, breadth$_{unprotected}$ = 810) and 28% higher after immunization (breadth$_{protected}$ = 943, breadth$_{unprotected}$ = 738), as antibody breadth was highly over-dispersed and sample size in the protected group small we found that differences in the means are not significant for either time points (logistic regression p-values of 0.6 and 0.3, respectively) (*Figure 6*). Furthermore, there was also limited discrimination by protection status in antibody breadth at both time points via receiver operating characteristics (ROC) analysis (area under the ROC curve: AUC$_{pre-immunization}$ = 0.64, Wilcoxon rank sum test $W$ = 60, p-value=0.35; AUC$_{post-immunization}$ = 0.73, Wilcoxon rank sum test $W$ = 44, p-value=0.1).

Finally, in order to identify antibodies consistently present in the samples of the protected individuals, we defined common antigens to a group as antigens which are reactive in at least 80% of the samples for each of the groups (i.e. considering the signal intensity of a given antigen as a binary outcome, either reactive or non-reactive). Common antigens in the protected group (reactive in at least 4 out of the five samples) were higher than in the unprotected groups (reactive in at least 27 out of the 33 samples) both 2 weeks after last immunization (383 common antigens in the protected group versus 58 in the unprotected group, 2-sample test for equality of proportions with continuity correction p-value<2.2E-16, $N$ = 2804) (*Figure 7b*), and at baseline (350 reactive antigens in protected group versus 62 in unprotected group, p<2.2E-16) (*Figure 7a*). Both at baseline and 2 weeks after immunization, almost all reactive antigens in common in the unprotected group were also present in the protected group (60 out of 62 and 56 out of 58 antigens at baseline and after immunization, respectively) (*Figure 7*). The trend for higher common antigens in the protected group compared to the unprotected group was also noticeable for different thresholds, comparing antigens reactive in at least 60% or in 100% of the samples in a given group (*Figure 7—figure supplement 1*). Given the large difference in sample sizes between the protected (n = 5) and unprotected (n = 33) groups, bootstrap samples with n = 5 samples in each group were repeatedly drawn

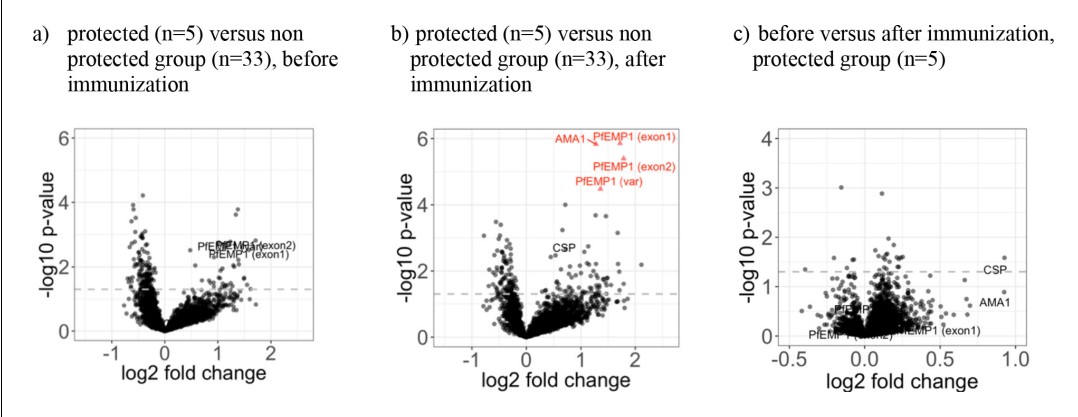

**Figure 5.** The five volunteers protected against homologous CHMI showed higher recognition of four distinct proteins after immunization. The mean fold change between antigen reactivity in the protected (n = 5) and the non-protected (n = 33) individuals from groups 2 and 3 are represented in volcano plots (**a**) for baseline and (**b**) after PfSPZ vaccination, plotted against the inverse $\log_{10}$ t-test p-value. In the protected group (n = 5), the samples collected at baseline and two weeks past last vaccination were compared and the mean fold change of the increased immunogenicity is showed in (**c**). Red triangles represent antigens with significant differences in antibody levels between protected and non-protected volunteers after BH adjustment of p-values, although size effect measured by Cohen's distance remains low (see *Figure 5—figure supplement 1*). The dashed line represents the threshold of statistical significance for the unadjusted p=0.05.

The online version of this article includes the following source data and figure supplement(s) for figure 5:

**Source data 1.** Source data for plot **a**.
**Source data 2.** Source data for plot **b**.
**Source data 3.** Source data for plot **c**.
**Figure supplement 1.** The increased recognition of four distinct proteins in the protected group show small effect size.
**Figure supplement 2.** No antigens show a significant differential antigen reactivity between the protected and unprotected group.
**Figure supplement 2—source data 1.** Source data for plot.
**Figure supplement 3.** Multiple sequence alignment of four PfEMP1 protein fragments.

(repeated 1000 times), with replacement. Consistent with previous analyses above, we find a higher number of common antigens in the protected compared to the unprotected group, although uncertainty due to small sample size is inevitable (*Figure 7—figure supplement 1*). Taken together this analysis suggests a higher number of commonly recognized antigens in the protected individuals after immunization and also at baseline, but the findings are limited by a small sample size in the protected group. A list of the commonly reactive antigens and antigens with increased reactivity levels following immunization per groups can be found in the *Figure 7—source data 1*.

To understand the biological function of proteins reactive in protected volunteers, we used Deep-Loc subcellular localization prediction, Pfam protein family prediction (*El-Gebali et al., 2019*), and gene ontology prediction available on Plasmodb.org (*Huntley et al., 2015*) and identified protein characteristics and distinct functional categories with higher representation in the protected volunteers. Predicted cell membrane proteins were more broadly recognized in the five protected volunteers at baseline (63 vs. 10 proteins, 2-proportions Z-test p-value=2.53E-9) and post-immunization (70 vs. 5 proteins, p-value=4.11E-13), but so were intracellular proteins localized to the cytoplasm, endoplasmic reticulum, Golgi apparatus, mitochondrion and nucleus (all p-values<0.05) (*Table 1*). Gene and protein families present in both protected and non-protected groups at both time points included integral components of membranes, host cell plasma membranes and infected host cell surface knobs, signal receptor activity and cell adhesion molecule binding, pathogenesis, cell-cell adhesion, antigenic variation and cytoadherence to the microvasculature, and PfEMP1-related families (see *Supplementary file 1*) . Gene and protein families uniquely reactive in the protected volunteer group at either or both time points included Mauer's cleft, host cell surface receptor binding, regulation of immune response, the Rifin protein family, a head domain of trimeric autotransporter adhesins (TAAs) family that acts as virulence factors for Gram-negative bacteria and have a head-stalk-anchor structure, a procyclic acidic repetitive protein family that was identified as abundant surface

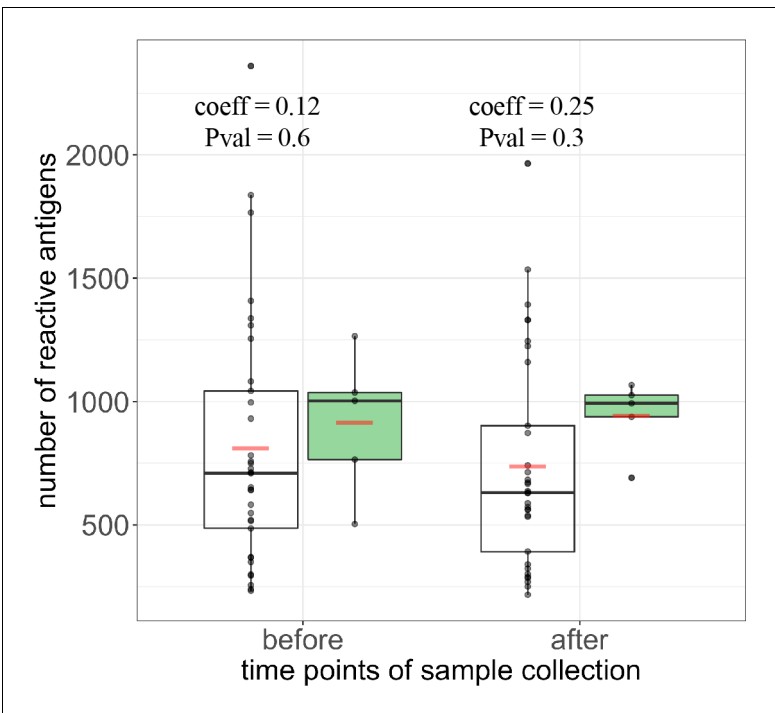

**Figure 6.** Breadth and magnitude of Pf-specific humoral immunity in protected and unprotected individuals. (**a**) Breadth counts of all PfSPZ vaccinees grouped by protection status following CHMI, with protected group in green and non-protected volunteers depicted in white, before and after immunization. The coefficient estimate with corresponding p-values from the negative binonial test is indicated for each time point. Boxplots show median, interquartile range (IQR) and 1.5xIQR limits and red bars represent the mean.

proteins in *Trypanosoma brucei*, and an N-terminal PRP1 splicing factor family involved in mRNA splicing (*Figure 7—source data 1*).

## Discussion

We provide first time data on the proteome-wide antibody profiling study for malaria endemic populations enrolled in a PfSPZ Vaccine phase Ib trial with homologous CHMI. Our analysis provides not only insights to PfSPZ vaccine induced humoral immune response and potential association with protection, but also a comprehensive view of underlying naturally acquired immunity in healthy Tanzanian adult men. The protein microarray used in this study considered a near full-proteome coverage (*Aurrecoechea et al., 2009*), contained full length or fragmented proteins representing 4805 (91%) of the approximately 5400 protein-coding genes in the *P. falciparum* 3D7 genome, the remaining 3D7 genes being tiny (<150 bp), challenging to clone or express or non-protein-coding genes. We found that a large proportion of the proteome was immunogenic in this study population, with personalized profiles detected by the protein microarray only moderately altered in response to PfSPZ Vaccine immunization. The moderate number of protected individuals (compared to volunteer infection studies in naïve populations), the heterogeneous immune fingerprint in the study population, and the potentially higher breadth of humoral immune response in the protected individuals highlight a complex picture to understand if humoral immune mechanisms lead to protection. Importantly, this study suggests that whole sporozoite vaccines predominantly boost pre-existing immunity of pre-exposed adults as a result of the natural imprinting of individual immune responses. Additionally, the moderate number of protected individuals seem to indicate that boosting pre-existing humoral response might not be sufficient to induce sterile protection against infection. If confirmed, these findings have implications for the role of such vaccines in endemic populations and thus prompts the need for studies to define appropriate target ages for immunization in settings of variable levels of pre-exposure.

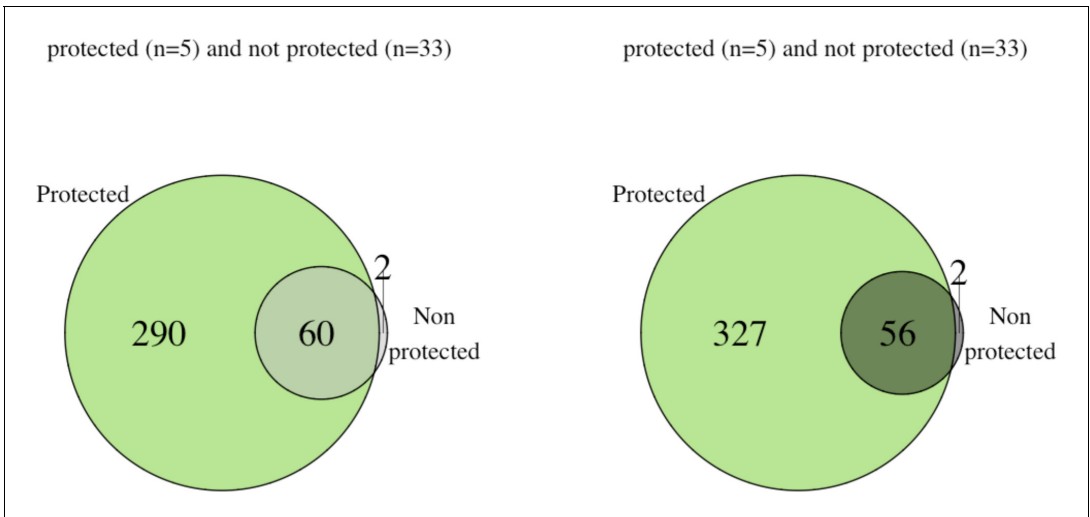

**Figure 7.** Protected individuals showed higher numbers of reactive antigens compared to the non protected group. The number of antigens that were reactive in at least 80% of the individual in each group are represented for (a) baseline for the protected group (green, n = 5) and the unprotected group (grey, n = 33) and for (b) after immunization for the protected group (green) and unprotected group (grey).
The online version of this article includes the following source data and figure supplement(s) for figure 7:

**Source data 1.** Table of commonly recognized antigens .
**Figure supplement 1.** The number of commonly recognized antigens, per threshold.

The humoral immune response in the malaria pre-exposed study population was broad, with 2239 unique proteins considered reactive among the serum samples tested. This large number of immunoreactive targets is surprising, especially considering that the majority of targets are predicted to be localized intracellularly. This is in stark contrast to a recent panproteome-wide analysis of antibody-binding targets for *Streptococcus pneumoniae* proteins, which identified a more restricted set of a few hundred antigens and a clear association of immunoreactivity with cell surface localized, albeit functionally diverse, pneumococcal proteins (*Croucher et al., 2017*). The tendency of malaria antigens to span a more comprehensive set of cellular compartments is likely due to the increased complexity of the life cycle and chronicity of infection. In hepatocytes and erythrocytes, the biological processes of schizogony occur in parasitophorous vacuoles culminating in the destruction of the host cell and likely release of numerous abundant parasite proteins into the immune system (*Cowman et al., 2016*). We speculate this results in substantial immune responses to intracellular parasite proteins and a significant dedication of host immune resources that are potentially functionally irrelevant for control of infection. However, the potential for these proteins to initiate cascades of cellular responses that aid in parasite clearance should not be discarded and could be the focus of future high throughput cellular antigen discovery.

Volunteers had a personalized antibody profile at baseline, and this profile remained at the same level of individual complexity after repeated PfSPZ Vaccine inoculation. Most individuals had an extensive breadth of immune response, but the number of reactive antigens that the samples have in common was low (around 10%). This implies that the majority of reactive antigens are distinctly recognized at the individual level and confirming uniqueness of reactivity profiles across individuals, further suggesting clonal imprinting is occurring in malaria immunity. Such imprinting is consistent with antigen recognition being less variable in adult populations experiencing seasonal exposure compared to children (*Taylor et al., 1996*), with adults consistently either seropositive or seronegative for a specific antigen throughout the transmission season. Of note, antibody 'fingerprints' were recently reported in volunteers of a *S. pneumoniae* whole cell vaccine trial, whereby the overall antibody profile remained consistent even following multiple exposures to killed pneumococci (*Campo et al., 2018*). A recent human cytomegalovirus (HCMV) vaccine study also found immune fingerprints where the vaccine boosted pre-existing immune responses implying a substantial effect of prior natural infection on vaccine induced immune responses for HCMV (*Baraniak et al., 2019*). These examples along with a review on this possible mechanism of fingerprinting in nine pathogens,

including Pf (*Vatti et al., 2017*), indicate that the phenomenon of 'original antigenic sin' which is known in the context of seasonal influenza vaccine goes beyond virus diseases. Our findings add to the increasing knowledge that – across the taxa - the personalized immune and metabolic status and history of pathogen exposure may affect vaccine take and the potential to elicit high levels of protection in vaccinated populations (*Vatti et al., 2017*; *Tsang et al., 2020*; *Hill et al., 2020*; *Baraniak et al., 2019*).

A consistent picture is emerging that pre-exposure limits the magnitude of vaccine induced responses. The same immunization regimen as group 3 in our Tanzanian study was administered to Malian adults, with antibody responses to PfCSP 6 to 7-fold lower in Malians than in the Tanzanians (*Jongo et al., 2018*; *Sissoko et al., 2017*). In contrast, higher anti-PfCSP levels were found following immunization in naïve U.S. vaccinees measured by same ELISA (*Epstein et al., 2017*; *Seder et al., 2013*) as well as in unpublished proteome-wide analysis (*Campo et al., 2018*), and in proteome-wide responses in European volunteers given chemo-attenuated sporozoites (PfSPZ-CVac) (*Mordmüller et al., 2017*). Studies of immunity induced after RTS,S vaccination have shown that anti-PfCSP titres are lower in Kenyan adults compared to naïve U.S. subjects, and a recent review on the immune response in RTS,S trials led to the hypothesis that pre-exposure might generate natural imprinting (*Vekemans, 2017*). It is possible that, as with PfCSP antibodies, the magnitude of vaccine-induced responses to other parasite antigens were lower in Tanzanian vaccinees than their U.S. and E.U. counterparts (*Kester et al., 2009*; *Polhemus et al., 2009*).

Although we cannot conclude on correlates of protection from this study, we did find an extensive breadth of humoral immune response in protected individuals indicating an underlying immune profile of wider antigen recognition compared to non-protected. Breadth is known to be associated with reduced risk of malaria (*Osier et al., 2008*; *Daou et al., 2015*), and anti-malaria protection likely to be the sum of protective immunity across different antigens, a concept known as the threshold of immune response (*Doolan and Hoffman, 1997*). Several recent protein microarrays studies found a range of antigens that were associated with protection (*Crompton et al., 2010*; *Trieu et al., 2011*; *Dent et al., 2015*). However, given the similar breadth profiles in the placebo group before and after placebo inoculation and the lack of any protected individuals in that group, it is more likely that the tendency for protected individuals to have higher antibody breadth represents a predisposition toward more effective PfSPZ Vaccine 'take', indicating that greater breadth prior to immunization may positively impact vaccination outcome. IgG specific to PfAMA1 and 3 variants of PfEMP1 were higher before CHMI in protected versus non-protected volunteers, but given the heterogeneous immune responses across volunteers, and the limited number of protected individuals, these antigens are not proposed as the mechanism for PfSPZ Vaccine induced immunity from this study. The three identified PfEMP1 antigens had higher levels in the protected individuals at baseline, indicating that higher levels of naturally primed, pre-existing PfEMP1 antibodies might be able to cross-react with other PfEMP1 proteins during CHMI.

We further found recognition of many more protein families in the protected group, including cell membrane proteins and numerous intracellular proteins. Protein families and functional categories uniquely identified in the protected group included the sporozoite protein SSP2/TRAP, a protein required for invasion of hepatocytes (*Mota and Rodriguez, 2004*). Interestingly, SSP2/TRAP antibodies were also detected in 26% of U.S. volunteers immunized with the same regimen of PfSPZ Vaccine as group 3 (*Epstein et al., 2017*). Additionally, the Rifin family, which are variant surface antigens exported to the infected erythrocyte surface and implicated in cytoadherence to the microvasculature and severe malaria (*Goel et al., 2015*) and the Mauer's cleft, which is involved in trafficking variant surface antigens, including Rifins and PfEMP1s, to the infected erythrocyte surface (*Mundwiler-Pachlatko and Beck, 2013*). While in many cases the magnitude of individual antibody responses was not significantly associated with protection, the analysis of protein families and functional categories can shed light on the types of antigens that may be targeted by a broad repertoire of antibodies conferring protection.

Given the small number of protected individual (n = 5) and the highly heterogeneous immune responses among the volunteers, results on potential association with protection of both individual antigens and breadth of humoral response in this study require further evaluation. Furthermore, four volunteers were protected in the highest immunization dose group (total of 1.35 million radiation attenuated sporozoites). Of particular note, the overall declining trend in antibody levels post-vaccination in both the lower dose and placebo groups was not evident in the higher dose group. Taken

together, this indicates that vaccination with a higher dose induced a maintained immune status over time, and thus increasing doses in African pre-exposed volunteers may rescue vaccine immunogenicity and an active memory B cell pool. Larger trials will be required to confirm or reject that higher immunization doses lead to increased protection level in pre-exposed adults, with recent studies indicating a four-fold increase in immunization dose did not increase efficacy compared to a two-fold increase (*Jongo et al., 2019*). Our results pertain to adults only and, consistent with the theory that high malaria pre-exposure reduces vaccine induced immune response, a PfSPZ Vaccine may well induce higher immune responses in children of endemic areas compared to adults (*Jongo et al., 2019*). Further studies are needed to understand the level of protection a PfSPZ Vaccine in all age groups and consequently, the likely vaccine efficacy achieved in a mass vaccination strategy.

Both PfSPZ Vaccine induced humoral and cellular immune response have been observed and associated with protection in studies in naïve volunteers, but an association of either or both of these responses with protection in pre-exposed populations is unclear. Previously, the primary immune response to PfSPZ immunization was thought to be cellular based (*Epstein et al., 2011*) including that increased cellular immune response following PfSPZ immunization observed in malaria naïve individuals (*Ishizuka et al., 2016*). However, these responses measured in peripheral blood are not correlated with protection in pre-exposed immunized adults assessed by CHMI (*Jongo et al., 2018*; *Jongo et al., 2019*), likely due to protective cellular immune effector cells residing in the liver (*Ishizuka et al., 2016*). Furthermore, there is accumulating evidence that PfSPZ Vaccine induces or boosts humoral immunity to a surprisingly limited number of antigens in pre-exposed adults, with no correlation between anti-CSP antibody titres and CHMI protection (*Jongo et al., 2018*; *Jongo et al., 2019*; *Sissoko et al., 2017*). This is despite previous reports that antibody responses are induced in US naïves (*Ishizuka et al., 2016*) and functional antibodies have been isolated from PfSPZ studies in Tanzania (*Tan et al., 2018*; *Zenklusen et al., 2018*).

Although we undertook an unbiased analysis approach, the use of microarrays comes with limitations. Firstly, false-positive discovery adjustments for protein microarray analysis potentially underestimate the association of antigens, especially where differences are subtle and heterogeneity between the samples is high. Secondly, as for most protein microarray studies, sample sizes replicates were not preformed. Nevertheless, the immune fingerprint was similar for both samples before and after immunization for each volunteer. As the array experiment was designed to balance samples across experimental nuisance factors, such as study day and sample order, it is unlikely that variability observed between volunteers is attributable to non-reproducibility of the experiment. Thirdly, the proteins and protein fragments of the microarrays are produced in a cell-free environment resulting in several epitopes lacking post-translational modifications and potentially folded into unnatural conformations that therefore might not be bound by specific serum antibodies, resulting in false-negative results (*Doolan et al., 2008*). For this reason, this technology has been described as a 'rule-in' and not a 'rule-out' method (*Stone et al., 2018*). Lastly, the paucity of information on the many functionally uncharacterized proteins present in the 5400 gene Pf proteome led to reliance on primarily *in silico* sequence prediction software to classify protein functional categories. Nevertheless, despite malaria-specific inaccuracies (details in Materials and methods), the DeepLoc analysis used in this study alongside gene ontology and protein family analysis provides valuable insight into overall distributions of the thousands of reactive antigens and those most recognized in protected individuals.

As information on the volunteers was limited, and as per design the study selected an apparently homogeneous population, it was impossible in the current analysis to examine associations of immune responses to different parasite or host factors. Thus, we cannot exclude associations between breadth of humoral immune responses, geographic location, immunogenetic background, transmission intensity, or other factors. Given the complexity and personalized immune response, we expect that much larger sample size, or a population meta-analysis, would be needed to identify any pattern of humoral response associated with host and parasite factors.

Further reduction and eventual elimination of malaria requires significant investment and research and development of new tools, including vaccines or other immune therapies (*Greenwood, 2008*). Our proteome-wide analysis indicates the breadth of antibody repertoire to Pf malaria is extensive and highly variable between individuals who are pre-exposed. Our findings and those from other PfSPZ Vaccine trials in Africa are subject to confirmation with future research studies before any

guidance can be made on the impact of pre-exposure on PfSPZ Vaccine efficacy and the implications for vaccination strategies. Nevertheless, we suggest, if these findings are confirmed, that the underlying, but difficult to assess, level of pre-exposure and resulting immune imprinting at the individual level may result in a more heterogeneous response to PfSPZ Vaccine. If personalized responses occur in pre-exposed individuals, then populations from different endemic regions cannot be considered homogeneous, and this will impact likely vaccination strategies. Similar to recent evidence for other pathogens (*Baraniak et al., 2019*; *Campo et al., 2018*; *Vatti et al., 2017*; *Tsang et al., 2020*), the potential impact of natural imprinting of humoral immune response in malaria deserves further investigation. Timing and duration of imprinting (infant age, childhood or throughout adulthood), as well as the role of co-infections and other yet to be identified host or environmental factors, are unknown. Without further fundamental studies, additional hurdles for future vaccine trials remain in regards to the validity of extrapolating vaccine outcomes from trials in naïve cohorts to pre-exposed populations and different age groups.

# Materials and methods

## Ethic statement

The study was approved by institutional review boards (IRBs) of the IHI (Ref. No. IHI/IRB/No:02–2014), the National Institute for Medical Research Tanzania (NIMR/HQ/R.8a/Vol.IX/1691), the Ethikkommission Nordwest-und Zentralschweiz, Basel, Switzerland (reference number 261/13), and by the Tanzania Food and Drug Authority (Ref. No.TFDA 13/CTR/0003); registered at Clinical Trials.gov (NCT02132299); and conducted under U.S. FDA IND 14826.

## Study design of the original trial

The design and outcome of the clinical study is described in detail in *Jongo et al., 2018*. Briefly, volunteers were immunized five times with a lower ($1.35 \times 10^5$) or a higher dose ($2.7 \times 10^5$) of PfSPZ Vaccine by direct venous inoculation (DVI) at 4 week intervals for the first four vaccinations followed by a last booster with PfSPZ Vaccine after 8 weeks. After immunization, volunteers underwent CHMI either 3 weeks after last immunization, 24 weeks after last immunization, or both using 3200 non-attenuated aseptic, purified, cryopreserved, infectious PfSPZ of PfSPZ Challenge administered by DVI. Serum samples for microarray analysis were collected at baseline (before immunization) and 2 weeks after last immunization in the individuals who underwent CHMI at 3 weeks after last immunization (*Figure 1*).

The 36 volunteers were healthy, adult males between 18–35 years old, with no parasitemia at the start of the study (measured by TBS and antibodies to PfEXP1 by ELISA), no history of malaria episodes over the last 5 years, and no parasitemia before CHMI (measured by TBS and qPCR) (*Jongo et al., 2018*). They were all students in Dar Es Salaam at the time of the study, however home town or travel history was not specified; thus, history of geographic exposure is not known.

## Protein array chip design

The protein microarray used in this study was produced by Antigen Discovery, Inc (ADI) and encompasses 7455 full-length or fragmented Pf proteins representing 4805 protein-coding genes and covering 91% of the proteome (*Mordmüller et al., 2017*). As previously described (*Felgner et al., 2013*) proteins were expressed from a library of Pf partial or complete open reading frames (ORFs) cloned into a T7 expression vector pXI using an in vitro transcription and translation (IVTT) system, the *Escherichia coli* cell-free Rapid Translation System (RTS) kit (5 Prime). This library was created via an *in-vivo* recombination cloning process with PCR-amplified Pf ORFs, and a complementary linearized expressed vector transformed into chemically competent *E. coli* was amplified by PCR and cloned into pXI vector using a high-throughput PCR recombination cloning method (*Davies et al., 2005*). Each expressed protein includes a 5′ polyhistidine (HIS) epitope and 3′ haemagglutinin (HA) epitope. Proteins were expressed according to manufacturer's instructions and then translated proteins were printed onto nitrocellulose-coated glass AVID slides (Grace Bio-Labs) using an Omni Grid Accent robotic microarray printer (Digilabs, Inc). Quality checks of the microarray chip printing and protein expression were performed by probing random slides with anti-HIS and anti-HA monoclonal antibodies with fluorescent labelling. In addition to the 7,455 Pf peptide fragments, each microarray

chip contained 302 IgG positive control spots as an assay control and 192 in vitro Transcription and Translation (IVTT) control spots (IVTT reactions with no Pf ORFs) as a normalization factor. All the spotted proteins were printed in three replicated pads per slide to accommodate one sample per pad. The experiment included two chips that made up the full proteome microarray, and samples were probed on each chip. Due to cost constraints, we did not replicate the experiment. Prior to probing samples, a balanced array experimental design was generated to mitigate nuisance factors, including pad position and day that sample was assayed, against sample grouping factors such as time point, dosing group and protection status. Sample balancing factors were provided as blinded, coded variables by Sanaria, Inc to ADI and unblinded following data acquisition.

### Sample probing

Sample probing has been previously described elsewhere (*Campo et al., 2015*; *Mordmüller et al., 2017*). Briefly, serum samples were diluted 1:100 in a 3 mg ml−1 *E. coli* lysate solution in protein arraying buffer (Maine Manufacturing) and incubated at room temperature for 30 min. Chips were rehydrated in blocking buffer for 30 min. Blocking buffer was removed, and chips were probed with serum samples by incubating in sealed, fitted slide chambers to ensure no cross-contamination of sample between pads. Chips were incubated overnight at 4°C with agitation. Chips were washed five times with TBS-0.05% Tween 20, followed by incubation with biotin-conjugated goat anti-human IgG (Jackson ImmunoResearch) diluted 1:200 in blocking buffer at room temperature. Chips were washed three times with TBS-0.05% Tween 20, followed by incubation with streptavidin-conjugated SureLight P-3 (Columbia Biosciences) at room temperature protected from light. Chips were washed three times with TBS-0.05% Tween 20, three times with TBS, and once with water. Chips were air dried by centrifugation at 1000 g for 4 min and scanned on a GenePix 4300A High-Resolution microarray scanner (Molecular Devices), and spot and background intensities were measured using an annotated grid file (.GAL). Data adjusted for local background by subtraction were exported to Microsoft Excel as CSV files and subsequently imported into R (*R Development Core Team, 2015*) where all subsequent data processing occured.

### Protein array data processing

Signal intensities were transformed by base two logarithm, and the median of IVTT control spots for each sample was subtracted from the sample-specific IVTT Pf antigen signals, a method that has been used previously in protein microarray analysis (*Mordmüller et al., 2017*; *Felgner et al., 2013*). A seropositive threshold was defined as two times IVTT control signals, or 1.0 on the $\log_2$ scale. A value of 0.0 +/- 1 represents signal intensities that are equivalent to the background. Values below −2, representing less than 0.25 times the median IVTT control signals were adjusted to −2. This affected 1639 of the 685,860 signals included in the complete dataset and 42 of the 257,968 signals included in the set of reactive antigens. Reactive antigens were defined as proteins that were seropositive in at least 10% of the study population at one or more time points. High level group reactivity was defined as 80% seropositivity to one probe in vaccinees who received either the lower (group 2) or higher (group 3) doses of PfSPZ Vaccine.

### Analysis

To visualize the high dimensional dataset of the microarray spots and understand potential patterns or clustering of the samples, t-SNE analysis (*Maaten and Hinton, 2008*) was used with a perplexity value of 30 and with 10,000 iterations. The t-SNE algorithm was applied the 92 samples of the entire dataset of the 2804 $\log_2$-transformed signal intensities or for the subset of 441 reactive proteins fragments predicted to be expressed at the sporozoite stage (*Florens et al., 2002*). The breadth of immune response for each individual was defined as the total number of positive reactive antigens for each serum draw. Breadth data were identified as over-dispersed after observing that the variance was greater than the means. Therefore, breadth between different groups was compared using negative binomial regression. The frequencies of reactive antigens summed into subcellular localization categories for each group were tested using 2-propotions Z-test and p-values adjusted using the BH method. Gene Ontology (GO) annotation for each protein was retrieved from PlasmoDB.org. Protein families were queried using amino acid sequences for each protein using the Pfam database

(*El-Gebali et al., 2019*). Fisher's exact tests were used to assess reactivity of each GO category of Pfam functional group, followed by p-value adjustment using the BH method.

The mean reactivity of each antigen per study group (control, group 2 and group 3) 2 weeks after immunization was compared with baseline levels (pre-immunization), using the paired t-test and by adjusting for false discovery rates (FDR) with the Benjamini-Hochberg method (BH) (*Benjamini and Hochberg, 1995*). One individual in group 2 who received 4 instead of 5 doses of immunization was excluded from this analysis. 'Delta' was defined as the fold change in antibody reactivity for each antigen before and after immunization. The average delta was compared between the vaccinated and the control groups using the unpaired t-test, and the resulting p-values were adjusted with the BH method. Because of the heteroscedastic nature of the normalized $\log_2$ signal intensities (*Figure 4—figure supplement 3*) we preferred the ordinary t-test over the empirical Bayes test (eBayes) (*Smyth, 2004*), which is sometimes used to compare signal intensities in microarray experiments. Due to the large number of positive probe signals and changes between time points, the eBayes could estimate a prior distribution that is over-dispersed relative to the paired t-test resulting in reduced power to detect changes in outlier responses, whereas eBayes may be more suitable for full proteome microarray studies with more restricted immunoreactivity profiles (*Campo et al., 2018*).

Common antigens to a group were defined as antigen which are reactive in at least 60%, 80% or 100% of the samples of the group and differences in frequencies of commonly recognized antigens between groups were assessed with a 2-sample test for equality of proportions with continuity correction. To account for the large difference in sample sizes between the protected (n = 5) and unprotected (n = 33) groups, bootstrap samples with n = 5 samples in each group were repeatedly drawn (repeated 1000 times), with replacement.

Amino acid sequences for each protein were queried using the DeepLoc online program to predict subcellular localization of each protein (*Almagro Armenteros et al., 2017*). Algorithms for prediction of subcellular localization of eukaryotic parasite proteins have not been trained like they have for Gram negative and positive bacteria, mammalian, plant and fungal cells, and DeepLoc program was trained to the latest UniProt dataset that reported higher accuracy using primary sequence input and deep neural networks over methods reliant on homology (*Almagro Armenteros et al., 2017*). No subcellular localization categories exist in these algorithms for the specialized organelles of malaria, such as the rhoptries and micronemes, which exocytose internal parasite proteins to the surface of the parasite membrane, parasitophorous vacuoles or host cell cytoplasm and membrane. Thus, many proteins such as the majority of PfEMP1 variants were misclassified as localized to the endoplasmic reticulum, golgi apparatus, cytoplasm or nucleus. The parasite apicoplast is also absent from these algorithms. Training to these organelles can only be done with a substantial database of sequences with which to train models, which is currently in limited supply for eukaryotic parasites.

## Acknowledgements

We thank all the volunteers in the trial, and all the researchers and staff involved in the clinical trial. We also thank Nicholas J Croucher for kindly sharing his code on the t-SNE algorithm. Finally, we thank the anonymous reviewers for their many suggestions for improving this paper.

## Additional information

### Competing interests

Joseph J Campo, Timothy Q Le, Amit Oberai, Christopher Hung, Jozelyn V Pablo, Andy A Teng, Xiaowu Liang: is an employee of Antigen Discovery, Inc. B Kim Lee Sim, Stephen L Hoffman: is employed by Sanaria. Sanaria Inc manufactured PfSPZ Vaccine and PfSPZ Challenge. Thus, all authors associated with Sanaria have potential conflicts of interest. The other authors declare that no competing interests exist.

## Funding

| Funder | Grant reference number | Author |
|---|---|---|
| Schweizerischer Nationalfonds zur Förderung der Wissenschaftlichen Forschung | PP00P3_170702 | Melissa A Penny |

The funders had no role in study design, data collection and interpretation, or the decision to submit the work for publication.

## Author contributions

Flavia Camponovo, Software, Formal analysis, Visualization, Methodology, Writing - original draft, Writing - review and editing; Joseph J Campo, Conceptualization, Data curation, Formal analysis, Supervision, Investigation, Writing - original draft, Writing - review and editing, Performed the protein localization; Timothy Q Le, Data curation, Investigation, Perform microarray experiment; Amit Oberai, Investigation, Performed the protein localization predictions; Christopher Hung, Jozelyn V Pablo, Andy A Teng, Data curation, Investigation, Performed the protein microarray experiments; Xiaowu Liang, Data curation, Investigation; B Kim Lee Sim, Funding acquisition, Conducted and supervised pharmaceutical operations, vaccine shipments of investigational products, Sanaria Inc sponsored the BSPZV1 study; Said Jongo, Resources, Conducted and supervised the BSPZV1 clinical trial; Salim Abdulla, Resources, Conducted and supervised the BSPZV1; Marcel Tanner, Supervision, Writing - review and editing; Stephen L Hoffman, Funding acquisition, Writing - review and editing, Sponsored the BSPZV1 study; Claudia Daubenberger, Conceptualization, Resources, Supervision, Writing - original draft, Writing - review and editing, Conducted and supervised the BSPZV1 study; Melissa A Penny, Conceptualization, Formal analysis, Supervision, Funding acquisition, Validation, Visualization, Methodology, Writing - original draft, Project administration, Writing - review and editing

## Author ORCIDs

Flavia Camponovo [ID] https://orcid.org/0000-0002-9762-851X
Melissa A Penny [ID] https://orcid.org/0000-0002-4972-593X

## Decision letter and Author response

Decision letter https://doi.org/10.7554/eLife.53080.sa1
Author response https://doi.org/10.7554/eLife.53080.sa2

# Additional files

## Supplementary files

- Source data 1. Gene Ontology prediction for the molecular function of the Pf genes.
- Source data 2. Gene Ontology prediction for the cellular component of the Pf genes.
- Source data 3. Gene Ontology prediction for the biological process of the Pf genes.
- Source data 4. Pfam database for the prediction of protein families.
- Supplementary file 1. Gene and protein families present in the protected versus non protected groups. This table lists Pfam protein family prediction (*El-Gebali et al., 2019*), and gene ontology prediction available on Plasmodb.org (*Huntley et al., 2015*) and identified protein characteristics and distinct functional categories which were identified as being reactive in at least 80% of the protected or non protected group before and after immunization. Reactive proteins were associated to each group using the Fisher's exact test, and p value correct using the Benjamini-Hochberg method (BH) (*Benjamini and Hochberg, 1995*). Pfam and GO description were found in https://www.ebi.ac.uk/QuickGO/ and https://biocyc.org/ and https://www.ebi.ac.uk/QuickGO/ and https://biocyc.org/, respectively. See also *Source datas 1–4*.

- Transparent reporting form

## Data availability

All data analyzed during this study are included in the manuscript and supporting files, or cited accordingly when published elsewhere.

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
