## [Decision Letter]

**Acceptance summary:**

Attenuated *Plasmodium falciparum* sporozoites (PfSPZ) represent one of the potential whole organism vaccines against malaria. In this study conducted in Tanzania, adult males were immunized with PfSPZ and protection against malaria was assessed by exposure to infectious Pf sporozoites. Antibody responses measured prior and after vaccination suggest that the PfSPZ induced but a moderately increased response and that the pre-existing immune responses might have reduced vaccine induced reactivity.

**Decision letter after peer review:**

Thank you for sending your article entitled "Proteome-wide humoral immunity of immunized adult malaria pre-exposed volunteers reveals personalized antibody profiles" for peer review at *eLife*. Your article has been evaluated by three peer reviewers, and the evaluation has been overseen by a Reviewing Editor and Dominique Soldati-Favre as the Senior Editor.

On the basis of the reviews provided by the three reviewers, there is a consensus regarding critical issues that require your attention. It appears that for the exclusion criteria, you have relied on serological assays for determinations of previous exposure to malaria. A more sensitive assay, such as PCR, should be performed to exclude any possibility of a prior malaria at pre-vaccination. In addition, re-analysis of results shown in Figures 1-5 is warranted for transparency and clarity as well as for the conclusions that antibodies to AMA-1 and PfEMP1 are the key mediators of protection induced by PfSPZ immunization. In view of the possible modifications of your conclusions, please revise the Discussion section to reflect these changes.

Reviewer 1:

Materials and methods: It appears that subjects were excluded on the basis of serological test for anti PfEXP-1 antibodies. This may not have been sufficiently sensitive assay. Was PCR-based assay done to check for parasitemia? Please address this important issue as the presence of asexual parasites pre-vaccination could have influenced the baseline antibody responses recorded, as well as the participants' anti-vaccine responses.

Results: The text associated with results shown in Figure 3 is not informative, hence detailed and clearer explanation of panels C and D needs to be provided. Please revise the legends to remove errors. Legend for panel E is missing. Amongst the proteins reported to be recognized significantly higher post-immunization in the protected group vs. the non-protected group is AMA1 (subsection “Association with protection”, last paragraph). The authors need to better explain how this outcome reflects the listings in Figure 7—source data 1 in which AMA1 is documented as at the top of the list (recognized by 24/33 individuals?) of proteins 'Increased in non-Protected' as well as in the 'Group 3 increased in non-Protected' list? Was the documented increase in response to AMA1 in the protected group simply greater than that observed in the non-protected group? The authors need to clarify what appears to be an anomaly in the data provided.

Discussion: The issue of pre-existing antibody responses (immune imprinting) is of potential importance in the clinical development of such a vaccine, in that such testing necessarily involves trials in older (African) age groups. It may possibly be less of an issue in African children who are one of the stated target groups for the vaccine. On the other hand the authors have previously stated (Jongo et al., 2019) that the protection conferred by their vaccine is thought to depend primarily on induction of cell-mediated immunological responses. That being the case, it could be argued that the presence or absence of pre-existing antibody responses – that themselves are inevitably less well-developed in young African children – becomes a comparatively minor consideration in the context of implementation in a mass vaccination programme designed to halt transmission of the parasite. Or do the authors now think – on the basis of the results presented here – that antibody responses may indeed make a contribution to the protection induced by their vaccine?

From a general standpoint, the interpretation of the study findings as a whole perhaps merits some further thought. It seems somewhat surprising that the administration of a total of at least 675 000 live radiation-attenuated sporozoites is (i) required to induce sterile immunity, (ii) necessary to register some measurable change in anti-parasite antibody responses (i.e. an increase in % 'high magnitude' antigens recognized by protected individuals), and (iii) associated with a significant change, post-immunization and pre-challenge, with antibody responses to just 4 different individual proteins in the individuals shown to have sterile protection. Do these findings simply underline the authors' own stated hypothesis, namely that protection induced by the PfSPZ vaccine is associated primarily with a cellular rather than a humoral response? The results certainly do seem to indicate that the large numbers of irradiated sporozoites administered as a vaccine that terminate their development in the liver appear not to have the capacity to substantially affect a majority of the anti-parasite antibody responses measured in the study. The absence of statistically significant differences following appropriate adjustment attests to that.

In the context of the authors' stated concept of using PfSPZ Vaccine in mass vaccination programmes, the comparatively poor efficacy against homologous challenge infection in African adults shown by the study is not encouraging for such a vaccine if the intention is to deploy it at a broad community-wide scale. The authors should perhaps consider discussing further how they think modifications incorporating 'the role of vaccine dosing and regimen on vaccine-induced immunity' might contribute to improving the overall performance of the vaccine in older African age-groups. If homologous protection is already rather low, what are the implications for the levels of heterologous protection in the same group of individuals? Discussion of these aspects should also take into account their published findings (Jongo et al., 2019) indicating comparatively poor T cell-mediated responses in PfSPZ-vaccinated Tanzanian adults.

Reviewer 2:

Introduction: The Introduction clearly sets out the current state of development of RTS,S and Spz. The section on immune correlates would benefit from more clearly separating mechanistic studies in mice from human challenge work where in vivo correlates might be statistically inferred. Both are valuable and both have drawbacks. Before the last paragraph on the role of microarrays it would be useful to summarize the foregoing review of potential correlates – e.g. to state that lead candidate correlates/mechanisms appear to be a b and c or that the field continues to be wide open with no consistent findings. The Introduction would also benefit from a shorter version.

Materials and methods: Can the authors comment about the validation of the peptide structures? How many are properly folded/ have correct confirmation etc.?

Results: Figure 2. The decline in reactivity post vaccination looks striking in the control group and in group 2. Why is the volcano plot clearly asymmetrical? Is there a batch effect?

Likewise when breadth is said to increase in 6/18in group 2 this means it decreases in 12/18 and this seems to be borderline significant (p=0.03).

Figure 3. This is the first time protected vs. non-protected result appears in the analysis. Why not from the start and why not volcano plots as shown in Figure 1 for protected vs. non-protected?

Categorizing antigens as low/middle/high then showing numbers of antigens in box plots seems a very counter-intuitive way of presenting the results for multi-dimensional data, much less familiar than volcano plots. Can this be justified?

Figure 4. I do not understand how the t-SNE plots further probe the finding of breadth not increasing. This is claimed as the objective, but the two ideas are not further linked in the description of results.

I have questions on the sentence "The number of antigen features recognized in at least 80% of the protected group (4 out of 5 individuals) was higher than in the unprotected group (27 out of 33 individuals) 2 weeks after 281 last immunization (383 reactive antigens in the protected group versus 58 in the unprotected".

Why pick that 80% to test in comparison with the unprotected group? There is a very confident p value to get out of a comparison of 4 individuals with another group (2*10^-16). I cannot completely follow the description of the analysis, but I doubt that such a p value can be gained from a well justified significance test in such a small group. A more transparent and simple analysis is strongly suggested. A volcano plot showing distributions of p values and effect sizes would be better, and this has been missed out in Figures 1 and 2.

Antibodies to AMA1 are the "second most significant increases" but text doesn't cover details for the most significant (i.e. PfEMP1). The unadjusted p value is 0.02 and 0.0002 but adjusted p values are 0.42 and 0.26. This does not seem to fit with the previous text, and I feel the whole area of significance testing here needs more clarity.

Figure 5. Volcano plots are back, but again they are not showing a comparison with protected vs. non-protected but rather pre/post divided for protected vs. non-protected. The sets above of reactive antigens pre (a) and post (b) seem to imply that as much difference may have been in the pre-vaccination reactivity as in the post. Can this be clarified?

Taken together I do not accept the key conclusion in the Abstract that antibodies to PfEMP1 and AMA1 were the key mediators of protection here. I would also be puzzled from an a priori perspective as sub-unit vaccines based on AMA1 have not been highly protective whereas in contrast PfSPZ has, and it is hard to accept this is primarily due to AMA1 responses.

Reviewer 3:

Building on antibody focused assessments completed during the course of PfSPZ Vaccine studies in Tanzania, Camponovo et al. seek to describe changes in humoral immune responses pre/post PfSPZ Vaccine (2 different doses) or placebo via Pf protein microarrays. They present results that highlight the diversity of pre-existing responses in presumed malaria exposed population and lack of significant clear changes in those who are vaccinated; though some fold changes in AMA-1 and PfEMP1 were seen in those who were protected from homologous CHMI.

Results: This section seems a bit unclear in its focus and explanation. Based on the Introduction, I was expecting to see results showing change in responses trying to target vaccine specific responses and impact pre-existing immunity may have on these responses. Thus I was expecting clear figures with defining degree of pre-existing exposure (and clear explanation and rationale on how this is defined; was this only defined by array responses or was there another measure to attempt to capture an individual's pre-existing exposure – clinical or immunological; have these definitions been validated prior) and how PfSPZ vaccination changed this response compared to similar controls, within an individual, and even though the numbers are small for those protected, if that change in IgG profile(s) can be seen in protected versus non-protected. I believe Figure 3 panels are attempting to do this but find it hard to track clearly the story being told and the relevance on a bigger scale as it's discussed later on in the paper.

Discussion: This section goes into the individual complexity of the array signatures even prior to vaccination, but would be interested to see if there was further analysis into age, prior residence, occupation, known malaria exposure, etc impacted this initial signature rather than assuming all male participants coming into the study were of equal prior malaria exposure. The interesting impact from this research, I believe, is this assessment of possible pre-vaccination imprinting and what can we learn from this to improve vaccine strategies, including PfSPZ vaccine. I believe more should be focused on this given the results presented really had no clear post vaccination signature seen and protected vs. unprotected numbers were extremely small for wider conclusions on possible markers of protection. Exploring how different individuals with perhaps very similar exposure backgrounds come into a vaccine trial and if a vaccine can alter this pre-existing baseline to promote benefit is crucial to vaccine success and worth focusing more clearly on.

---

## [Author Response]

On the basis of the reviews provided by the three reviewers, there is a consensus regarding critical issues that require your attention. It appears that for the exclusion criteria, you have relied on serological assays for determinations of previous exposure to malaria. A more sensitive assay, such as PCR, should be performed to exclude any possibility of a prior malaria at pre-vaccination. In addition, re-analyses of results shown in Figures 1-5 is warranted for transparency and clarity as well as for the conclusions that antibodies to AMA-1 and PfEMP1 are the key mediators of protection induced by PfSPZ immunization. In view of the possible modifications of your conclusions, please revise the Discussion section to reflect these changes.Reviewer 1:Materials and methods: It appears that subjects were excluded on the basis of serological test for anti PfEXP-1 antibodies. This may not have been sufficiently sensitive assay. Was PCR-based assay done to check for parasitemia? Please address this important issues as the presence of asexual parasites pre-vaccination could have influenced the baseline antibody responses recorded, as well as the participants' anti-vaccine responses.

We thank reviewer 1 for their thoughtful and helpful comments. All points have influenced our updated manuscript.

We are confident that all volunteers were parasite free both at the start of the study and before CHMI. At enrolment, all volunteers were screened by thick blood smear (TBS) for malaria and only volunteers that tested negative were included (Supplementary Table 3 of Jongo et al., 2018). Volunteers were screened before CHMI (that is 21 days after fifth vaccination) by TBS and qPCR for malaria and all subjects were found to be negative using both detection methods (Jongo et al., 2018). The fact that none of the volunteers was detected during the vaccination period by TBS as malaria positive, that none of the volunteers developed malaria during the vaccination period, and that all volunteers were malaria negative (as per qPCR) before CHMI, strongly suggests that these volunteers were negative for malaria at enrolment. The details of exclusion criteria can be found in the clinical trial paper (Jongo et al., 2018), but the reviewer’s comment made it clear that this aspect is critical for the interpretation of our results and we have now added the following specifications in the manuscript:

In Results, section on Study volunteers and serum sampling:

*“*All volunteers included in the study had no parasitemia at the start of the study (measured by malaria thick blood smears (TBS)) and no parasitemia before CHMI (measured by TBS and the more sensitive qPCR) (Jongo et al., 2018). Additional exclusion criteria included history of malaria in the previous 5 years or antibodies to PfEXP1 by ELISA above a threshold level (Jongo et al., 2018) associated with recent infection by CHMI (Shekalaghe et al., 2014).”

And in the Materials and methods section “Study design of the original trial”:

“The 36 volunteers were healthy, adult males between 18-35 years old, with no parasitemia at the start of the study (measured by TBS and antibodies to PfEXP1 by ELISA), no history of malaria episodes over the last 5 years, and no parasitemia before CHMI (measured by TBS and qPCR) (Jongo et al., 2018). They were all students in Dar Es Salaam at the time of the study, however home town or travel history was not specified; thus, history of geographic exposure is not known.”

Results: The text associated with results shown in Figure 3 is not informative, hence detailed and clearer explanation of panels C and D needs to be provided. Please revise the legends to remove errors. Legend for panel E is missing.

Thank you, this is a very helpful comment as we realized that by modifying this figure and by restructuring the Results section we could improve the clarity of our Results section.

We have now changed the plots as follows.

i) Originally Figure 3A-B is now a separate figure (Figure 3B-C) and we also included breadth before immunization for each individual (Figure 3A);

ii) original Figure 3D-E on the magnitude of response have been removed, and

iii) original Figure 3C is now a separate figure (Figure 6).

We agree that the plot on magnitudes provided only weak additional information compared to plots of breadth of response, thus for clarity, we have removed these plots.

Amongst the proteins reported to be recognized significantly higher post-immunization in the protected group vs. the non-protected group is AMA1 (section “Association with protection”, last paragraph). The authors need to better explain how this outcome reflects the listings in Figure 7—source data 2 in which AMA1 is documented as at the top of the list (recognized by 24/33 individuals?) of proteins 'Increased in non-Protected' as well as in the 'Group 3 increased in non-Protected' list? Was the documented increase in response to AMA1 in the protected group simply greater than that observed in the non-protected group? The authors need to clarify what appears to be an anomaly in the data provided.

We thank reviewer 1 for this comment, which helped us to improve the clarity of our analysis. The Results section has been restructured in response to the overall comments of the reviewers. This restructure includes separating the Venn diagrams showing the commonly recognized antigens in 80% of the samples of a given group (updated Figure 7), and the volcano plots showing the mean difference in signal intensity between the protected and unprotected groups or the two sample time points (updated Figure 5). Separating those results and adding more details on these two sets of analysis, hopefully, clarifies the different outputs displayed in the Figure 7—source data 1 and the volcano plots.

Figure 7—source data 1 lists antigens that were reactive or where reactivity increased from baseline to after immunization, in at least 50% of the samples for a given group. The list indeed shows that 2 weeks after immunization PfAMA1 signals had increased compared to baseline levels in 4 out of 5 protected volunteers (80%) and in 24 out of 33 non-protected volunteers (72%). These lists do not indicate the magnitude of the increase or if the mean signal level before versus after immunization can be considered significantly different. This analysis was used to examine the number of commonly reactive antigens per group rather than the analysis of the reactivity levels of the individual antigens. The latter was performed by comparing the mean signals between different groups, evaluated using the student test statistics, and displayed through volcano plots. These volcano plots showed that the mean antigen reactivity for four proteins, including PfAMA1, was higher in the protected group compared to the non-protected group 2 weeks after immunization (Figure 5B in the new version of the manuscript). This is referred in the reviewer’s comment “Amongst the proteins reported to be recognized significantly higher post-immunization in the protected group vs. the non-protected group is PfAMA1.”

In the protected group, no antigens significantly increased in reactivity after immunization compared to baseline (Figure 5C). We did not look at the mean increase of PfAMA1 and other antigens in the unprotected group between before and after immunization, but given the number of protected (n=5) and unprotected (n=33) it would likely be similar to Figure 4B-C which shows the increase from baseline in all immunized samples (regardless protection) of group 2 and group 3. Although the mean log fold change of the signal of PfAMA1 was 0.3 in group 2 and 0.6 in group 3 (which are the second and third highest fold change in group 2 and group 3, respectively), it did not meet the significance threshold, and PfAMA1 is not highlighted in the plots. Delta, the differential increase before to after immunization, was not significantly higher in the protected compared to the unprotected group (Figure 5—figure supplement 2).

We hope that the restructuring of the Results as a whole makes it easier to follow the analysis, including clarifying this comment. Specifically, we included the following sentences:

At the beginning of the third paragraph of the section “Moderate increase in antigen recognition following immunization with PfSPZ Vaccine”:

“To identify potential differences in vaccine induced immunogenicity between protected and unprotected individuals, we compared the difference in the mean immunoreactivity (i.e. signal intensity) for each antigen at baseline and 2 weeks after immunization between protected (n=5) and unprotected (n=33) volunteers, and the difference in the mean immunogenicity of each antigen between baseline and post immunization time points in the protected group (Figure 5)”;

At the end of the 3rd paragraph of the section “Moderate increase in antigen recognition following immunization with PfSPZ Vaccine”:

“No antigen showed an increase in reactivity (delta) significantly higher in the protected group (Figure 5—figure supplement 2).”

At the beginning of the second paragraph of the section “Breadth of humoral immune response in protected individuals”:

“Finally, in order to identify antibodies consistently present in the samples of the protected individuals, we defined common antigens to a group as antigens which are reactive in at least 80% of the samples for each of the groups (i.e. considering the signal intensity of a given antigen as a binary outcome, either reactive or non-reactive).”

Discussion: The issue of pre-existing antibody responses (immune imprinting) is of potential importance in the clinical development of such a vaccine, in that such testing necessarily involves trials in older (African) age groups. It may possibly be less of an issue in African children who are one of the stated target groups for the vaccine. On the other hand the authors have previously stated (Jongo et al., 2019) that the protection conferred by their vaccine is thought to depend primarily on induction of cell-mediated immunological responses. That being the case, it could be argued that the presence or absence of pre-existing antibody responses – that themselves are inevitably less well-developed in young African children – becomes a comparatively minor consideration in the context of implementation in a mass vaccination programme designed to halt transmission of the parasite. Or do the authors now think – on the basis of the results presented here – that antibody responses may indeed make a contribution to the protection induced by their vaccine?

We thank the reviewer for that comment. We believe that to date, it remains unclear what immune mechanisms lead to protection following PfSPZ vaccine immunization, especially in pre-exposed population. To date cellular immune responses are thought to play a major role in protection, although cells located in the liver have not been measured in clinical studies and thus effective cellular immune response in the liver remains unclear. On the humoral side, functional antibodies have been isolated in both naïve and pre-exposed populations. Due to the small sample size of protected individuals, and the heterogeneous responses observed in this study we don’t think we can make strong statements on the role of humoral immune response for acquiring protection, but we think it should not be ruled out.

We have edited the Discussion section. In particular, to clarify current knowledge of cellular and humoral immune response, we added a ninth paragraph:

“Both PfSPZ Vaccine induced humoral and cellular immune response have been observed and associated with protection in studies in naïve volunteers, but an association of either or both of these responses to protection in pre-exposed populations is unclear. […] This is despite previous reports that antibody responses are induced in in naives (Ishizuka et al., 2016) and functional antibodies have been isolated from PfSPZ studies in Tanzania (Tan et al., 2018, Zenklusen et al., 2018).”

From a general standpoint, the interpretation of the study findings as a whole perhaps merits some further thought. It seems somewhat surprising that the administration of a total of at least 675 000 live radiation-attenuated sporozoites is (i) required to induce sterile immunity, (ii) necessary to register some measurable change in anti-parasite antibody responses (i.e. an increase in % 'high magnitude' antigens recognized by protected individuals), and (iii) associated with a significant change, post-immunization and pre-challenge, with antibody responses to just 4 different individual proteins in the individuals shown to have sterile protection. Do these findings simply underline the authors' own stated hypothesis, namely that protection induced by the PfSPZ vaccine is associated primarily with a cellular rather than a humoral response? The results certainly do seem to indicate that the large numbers of irradiated sporozoites administered as a vaccine that terminate their development in the liver appear not to have the capacity to substantially affect a majority of the anti-parasite antibody responses measured in the study. The absence of statistically significant differences following appropriate adjustment attests to that.

We thank the reviewer for this comment. We have completed major edits on the Discussion, including additional literature, to emphasize the primary outcome of this analysis, namely the effect of the natural imprinting on vaccine induced humoral immune response in pre-exposed adults. Whether or not humoral, cellular, or both immune responses are the main drivers of vaccine induced sterile immunity remains unknown, and hopefully, this is clarified with the additional section in the Discussion. We agree with the last comment of reviewer #2, but we suggest that the moderate change in the antibody immune response profile is due to natural imprinting, which is the main focus of the Discussion, mainly in the third and fourth paragraph.

We added a paragraph on the current knowledge about humoral and cellular immune response induced by PfSPZ Vaccine (see previous comment), and in the fourth paragraph of the Discussion we highlight that the vaccine induced humoral immune response are very different in malaria naïve and malaria exposed individuals, with naïve volunteers finding to have much higher PfSPZ Vaccine induced antibody levels. In addition, to highlight the reviewer’s comments (i), (ii) and (iii) in our Discussion, we have now added the following paragraph:

“Furthermore, 4 volunteers were protected in the highest immunization dose group (total of 1.35 million radiation attenuated sporozoites). […] Further, studies are needed to understand the level of protection a PfSPZ Vaccine in all age groups and consequently, the likely vaccine efficacy achieved in a mass vaccination strategy.”

In the context of the authors' stated concept of using PfSPZ Vaccine in mass vaccination programmes, the comparatively poor efficacy against homologous challenge infection in African adults shown by the study is not encouraging for such a vaccine if the intention is to deploy it at a broad community-wide scale. The authors should perhaps consider discussing further how they think modifications incorporating 'the role of vaccine dosing and regimen on vaccine-induced immunity' might contribute to improving the overall performance of the vaccine in older African age-groups. If homologous protection is already rather low, what are the implications for the levels of heterologous protection in the same group of individuals? Discussion of these aspects should also take into account their published findings (Jongo et al., 2019) indicating comparatively poor T cell-mediated responses in PfSPZ-vaccinated Tanzanian adults.

We agree with the reviewer’s comment that we have not emphasized strongly enough the implications of the findings for mass vaccination strategies. We don’t think our findings are sufficient to make recommendation either way on the use of such a vaccine in mass vaccination strategies, or suggest potential modifications that should be made. However, we suggest based on our findings that natural imprinting will be an additional challenge for vaccination in adults from African populations, and that further investigation is required to understand the direct implications. We hope that the significant edits to our Discussion better convey this message. In particular, to the reviewer’s comments, we have modified our concluding paragraph as follows:

*“*Our proteome-wide analysis indicates the breadth of antibody repertoire to Pf malaria is extensive and highly variable between individuals who are pre-exposed. […] Without further fundamental studies, additional hurdles for future vaccine trials remain in regards to the validity of extrapolating vaccine outcomes from trials in naïve cohorts to pre-exposed populations and different age groups.”

Reviewer 2:Introduction: The Introduction clearly sets out the current state of development of RTS,S and Spz. The section on immune correlates would benefit from more clearly separating mechanistic studies in mice from human challenge work where in vivo correlates might be statistically inferred. Both are valuable and both have drawbacks. Before the last paragraph on the role of microarrays it would be useful to summarize the foregoing review of potential correlates – e.g. to state that lead candidate correlates/mechanisms appear to be a b and c or that the field continues to be wide open with no consistent findings. The Introduction would also benefit from a shorter version.

We thank reviewer #2 for their thoughtful and helpful comments. These are constructive and helpful comments that provided significant inputs for improving the Introduction. With have shortened and restructured the Introduction for more clarity. We have separated in vitro from in vivo studies of protection correlates, as well as more clearly identified studies in mice, non-human primates, naïve humans and pre-exposed humans, as we agree that this would be easier to read. Our review of current knowledge of PfSPZ Vaccine induce immune response is now described in the fourth paragraph (cellular immune response with first studies in mice, followed by studies malaria naïve, and malaria exposed volunteers), fifth paragraph (humoral immune response), and sixth paragraph (studies on the functional role of antibodies).

To address the last comment, we have included the following sentence in the last paragraph of the Introduction to highlight that mechanisms leading to protection remain an open question:

“The main in-vitro, animal, and human studies described above suggest that both cellular immune response and antibody mediated immune response play a role in inducing protection. However, a complete understanding of the mechanisms of vaccine-induced protection against malaria infection, and its interplay with pre-built natural immune response in exposed populations, remains unknown”.

Materials and methods: Can the authors comment about the validation of the peptide structures? How many are properly folded/have correct confirmation etc.?

We thank the reviewer for pointing out this missing information in our Materials and methods. In the Discussion, we already highlighted the limitation of the protein microarray to ensure properly folded epitopes, and thus this technology being referred to as a “rule in” and not “rule out” method. But we have now added more specification in the Materials and methods:

“As previously described (Felgner et al., 2013) proteins were expressed from a library of Pf partial or complete open reading frames (ORFs) cloned into a T7 expression vector pXI using an in vitro transcription and translation (IVTT) system, the *Escherichia coli* cell-free Rapid Translation System (RTS) kit (5 Prime). […] Quality checks of the microarray chip printing and protein expression were performed by probing random slides with anti-HIS and anti-HA monoclonal antibodies with fluorescent labelling.”

Results: Figure 2. The decline in reactivity post vaccination looks striking in the control group and in group 2. Why is the volcano plot clearly asymmetrical? Is there a batch effect?Likewise when breadth is said to increase in 6/18in group 2 this means it decreases in 12/18 and this seems to be borderline significant (p=0.03).

We thank the reviewer for this comment. The overall decline in reactivity after immunization was surprising to us. We are confident that this is not due to experimental biases, as samples were balanced for group and time point factors across technical microarray factors using a block randomization design. Sample balancing factors were provided as blinded, coded variables by Sanaria, Inc to ADI and unblinded following data acquisition. This is specified in the Materials and methods section “Protein array chip design”, and highlighted this in the Results:

“During the period before and after vaccination, antibody breadth declined in many individuals in the control and immunized groups (note that samples were balanced for group and time point factors across technical microarray factors using a block randomization design, see Materials and methods)”.

We agree that the breadth of responses tends to decrease in the control group, and in group 2, and we measured the difference in breadth before and after vaccination with the inverted beta-binomial test for paired count data which resulted in an estimated fold change of -1.23, -1.16 and 1.03, and a p-value of 0.24, 0.03 and 0.43 for control, group 2, and group 3 respectively. Indeed, the fold change in group 2 has an estimated p-value of 0.03, nevertheless, breadth is very variable between individuals and in group 2 there seem to be 2 outliers with considerable decrease in breadth (which were highlighted in the t-SNE plots), thus we have not put too much emphasis on this finding. These two individuals are now also highlighted in the plots of breadths, and we have added as a last sentence on the results on breadth:

“Overall, there was no dramatic change in breadths between both time points, which aligns with the immune fingerprint analysis in Figure 2. There was a small decreased average breadth in group 2 driven by 2 of the three individuals whose samples did not cluster for immune-fingerprinting.”

Figure 3. This is the first time protected vs. non-protected result appears in the analysis. Why not from the start and why not volcano plots as shown in Figure 1 for protected vs. non-protected?Categorizing antigens as low/middle/high then showing numbers of antigens in box plots seems a very counter-intuitive way of presenting the results for multi-dimensional data, much less familiar than volcano plots. Can this be justified?

We thank reviewer 2 for this comment, which greatly helped us to structure our Results. We agree that the plot on magnitudes provided only weak additional information compared to plots of breadth of response, thus for clarity, we have removed these plots.

Results have been re-ordered, the protected individuals are now shown in earlier figures, and the volcano plots of the protected group (now Figure 5A-B, previously Figure 5C-D) now follow the volcano plots of the different dose groups (now Figure 4), for better clarity. In addition, we have moved the volcano plot showing the increase in immunogenicity from baseline to after immunization in the protected group to the main article (previously in the supplement) (Figure 5C). Finally, the analysis previously shown in Figure 3 has now been restructured as follows. Figure 3A-B is now a separate figure (Figure 3B-C) in which we also included breadth before immunization for each individual (Figure 3A), Figure 3D-E on the magnitude of response have been removed and Figure 3C is now a separate figure (Figure 6).

Figure 4. I do not understand how the t-SNE plots further probe the finding of breadth not increasing. This is claimed as the objective, but the two ideas are not further linked in the description of results.

We thank reviewer 2 for that comment. The Results restructuring puts the t-SNE results at the very beginning, followed by the results on breadth in each group, which hopefully makes it clearer. With this order, the t-SNE analysis is the first analysis, displaying the results of thousands of signal intensities onto two dimensions, and giving first indications that the patterns of humoral immune response are person specific and not drastically modified by immunization. The following analysis of breadth of humoral immune response provides additional information on the number of reactive antigens per individual and time point, and highlights the heterogeneity in breadth between individuals and a moderate change in breadths following immunization. We hope that the restructuring of our Results conveys this message better than our previous version.

Additionally, we gave more details on the t-SNE analysis at the beginning of the second paragraph of the section “Tanzanian male adults recognize a high diversity of Pf proteins”:

“First, we examined the antibody profiles for each volunteer individually and the results of the paired samples are presented in the heatmap (Figure 2A). […] The t-SNE algorithm estimates the probability distribution of neighbors around each point, i.e. it models the set of points which are closest to each point*.”*

I have questions on the sentence "The number of antigen features recognized in at least 80% of the protected group (4 out of 5 individuals) was higher than in the unprotected group (27 out of 33 individuals) 2 weeks after 281 last immunization (383 reactive antigens in the protected group versus 58 in the unprotected".Why pick that 80% to test in comparison with the unprotected group? There is a very confident p value to get out of a comparison of 4 individuals with another group (2*10^-16). I cannot completely follow the description of the analysis, but I doubt that such a p value can be gained from a well justified significance test in such a small group.

Yes, 80% has been taken as an arbitrary threshold, and we agree that this should be clear to the reader. We have now included results with a threshold of 60% and a threshold of 100% (antigen recognized in all sample of the group) as a supplementary figure (Figure 5—figure supplement 2). The test and corresponding p-value refer to comparing two proportions (383/2804 compared to 58/2804, 2804 being the total number of antigens) and thus does not include sample size. But we agree with reviewer 3 that this p-value could be misleading if we don’t address the issue of small sample size. We did specify in the text that these results are very sensitive to the small number of protected individuals. In addition, to highlight the uncertainty due to small sample size in the protected samples, we have now added an analysis taking a bootstrap sample of n=5 for both protected and unprotected group, repeating 1000 times (Figure 7—figure supplement 1).

We have also added the following text in the second paragraph of the section entitled “Breadth of humoral immune response in protected individuals”:

“The trend for higher common antigens in the protected group compared to the unprotected group was also noticeable for different thresholds, comparing antigens reactive in at least 60% or in 100% of the samples in a given group (Figure 7—figure supplement 1). […] Consistent with previous analyses above, we find a higher number of common antigens in the protected compared to the unprotected group, although uncertainty due to small sample size is inevitable (Figure 7—figure supplement 1).”

A more transparent and simple analysis is strongly suggested. A volcano plot showing distributions of p values and effect sizes would be better, and this has been missed out in Figures 1 and 2.

We agree with this comment, and as results have been re-ordered, the volcano plots of the protected group (now Figure 5A-B, previously Figure 5C-D) are now following the volcano plots of the different dose groups (now Figure 4), for better clarity. In addition, we have moved the volcano plot showing the increase in immunogenicity from baseline to after immunization in the protected group to the main article (previously in the supplement) (Figure 5C). We agree that it is crucial that all analysis is transparent on the uncertainty due to the low sample size in the protected group. Including effect sizes was a great suggestion, and we have added them as supplementary figures for the volcano plots for immunogenicity per dose group (Figure 4—figure supplement 1), the volcano plots comparing protected versus non-protected (Figure 5—figure supplement 1), and mentioned them where relevant in the Results (in legends of Figure 4 and 5).

Antibodies to AMA1 are the "second most significant increases" but text doesn't cover details for the most significant (i.e. PfEMP1). The unadjusted p value is 0.02 and 0.0002 but adjusted p values are 0.42 and 0.26. This does not seem to fit with the previous text, and I feel the whole area of significance testing here needs more clarity.

We thank reviewer 2 for pointing this sentence out. We agree that this sentence was not relevant in the paragraph, and we have removed it. Restructuring the Results moved this paragraph, and it has been modified to hopefully increase clarity.

Figure 5. Volcano plots are back, but again they are not showing a comparison with protected vs. non-protected but rather pre/post divided for protected vs. non-protected. The sets above of reactive antigens pre (a) and post (b) seem to imply that as much difference may have been in the pre-vaccination reactivity as in the post. Can this be clarified?

All reviewer’s comments made it clear that our Results should be heavily restructured, and restructuring has been described in the previous response to comments.

The volcano plots are now higher up in the Results, and the volcano plot showing the increase in reactivity in the protected group has now been added (Figure 5C). The difference in the signal intensities in protected versus non-protected group in the post-vaccination samples is higher and more significant than in the pre-vaccination samples. Nevertheless, the text corresponding to this analysis has been modified as it was not the intention to emphasize on this result. Adding the effect sizes was a great suggestion to highlight the uncertainty of those results due to small sample size.

The volcano plot of the difference in the antigen increase after immunization from baseline between the protected and unprotected group is kept as a supplementary plot, but mentioned as followed in the Results:

“No antigen showed an increase in reactivity (delta) significantly higher in the protected group (Figure 5—figure supplement 2).”

The corresponding paragraph, which is now the third paragraph of the section “Moderate increase in antigen recognition following immunization with PfSPZ Vaccine” has been shortened and modified. Among other modifications, we added the following sentence:

“Nevertheless, the sample size of the protected group is low (n=5), and considering the low effect size measured by Cohen’s distance (Figure 5—figure supplement 1), no strong argument for association with protection of these four antigens can be made from this analysis.”

Taken together I do not accept the key conclusion in the Abstract that antibodies to PfEMP1 and AMA1 were the key mediators of protection here. I would also be puzzled from an a priori perspective as sub-unit vaccines based on AMA1 have not been highly protective whereas in contrast PfSPZ has, and it is hard to accept this is primarily due to AMA1 responses.

This is a critical observation from the reviewer, the statement was too strong, and we thank the reviewer for being critical of this, as we do not wish to make too strong a statement based on our data. Additionally to the modifications made in the Results as mentioned above, we modified the Discussion to reduce the length of discussion on associations of protection and hopefully highlight the preliminary aspect of those results. The mention of the 4 identified antigens is now reduced to these two sentences:

“Although we cannot conclude on correlates of protection from this study, […] IgG specific to AMA1 and 3 variants of PfEMP1 were higher before CHMI in protected versus non-protected volunteers. The three identified PfEMP1 antigens had higher levels in the protected individuals at baseline, indicating that higher levels of naturally primed, pre-existing PfEMP1 antibodies might be able to cross-react with other PfEMP1 proteins during CHMI”.

Reviewer 3:[…] Results: This section seems a bit unclear in its focus and explanation. Based on the Introduction, I was expecting to see results showing change in responses trying to target vaccine specific responses and impact pre-existing immunity may have on these responses. Thus I was expecting clear figures with defining degree of pre-existing exposure (and clear explanation and rationale on how this is defined; was this only defined by array responses or was there another measure to attempt to capture an individual's pre-existing exposure – clinical or immunological; have these definitions been validated prior) and how PfSPZ vaccination changed this response compared to similar controls, within an individual, and even though the numbers are small for those protected, if that change in IgG profile(s) can be seen in protected versus non-protected. I believe Figure 3 panels are attempting to do this but find it hard to track clearly the story being told and the relevance on a bigger scale as it's discussed later on in the paper.

We thank the reviewer for this critical comment, which has helped us restructure our analysis. We have given more specification on the exclusion criteria of the clinical study, highlighting that volunteers should have no history of malaria over the 5 previous years and were parasite free at the start of the trial. The first section of the Results, “Study volunteers and serum sampling”, now includes these specification:

“All volunteers included in the study had no parasitemia at the start of the study (measured by thick blood smears TBS) and no parasitemia before CHMI (measured by TBS and qPCR) (Jongo et al., 2018). Additional exclusion criteria included history of malaria in the previous 5 years or antibodies to PfEXP1 by ELISA above a threshold level (Jongo et al., 2018) associated with recent infection by CHMI (Shekalaghe et al., 2014).”

Figure 3 has now been split and edited. The Results section now starts with the microarray heatmap and t-SNE plot showing the overall response at baseline and following immunization, highlighting the person specific immune fingerprint prior to immunization and the overall unchanged immune fingerprint post-immunization for individuals across the groups regardless of protection level. These results are followed by the plots of breadth of response, before going into changes in antibody levels following immunization and between protection level (volcano-plots) and finally differences between the protected and unprotected groups in the proteins recognized.

The new structure of the Results hopefully makes it easier for the reader to identify the preexposure levels earlier on in the analysis, and observe the changes in the humoral immune response induced by vaccination and differences per protection level.

Discussion: This section goes into the individual complexity of the array signatures even prior to vaccination, but would be interested to see if there was further analysis into age, prior residence, occupation, known malaria exposure, etc impacted this initial signature rather than assuming all male participants coming into the study were of equal prior malaria exposure.

We thank reviewer 3 and agree with this comment. We agree that other covariates such as age, malaria exposure or residence would be very interesting to investigate. We initially intended to extend our analysis to potential demographic covariates. However, we realized that given the available information on the individuals in the clinical trial, we were unable to make any distinctions between volunteers and decided to keep them as an a priori homogeneous population. We have expanded our Discussion to include this limitation as follows:

“As information on the volunteers was limited, and as per design the study selected an apparently homogeneous population, it was impossible in the current analysis to examine associations of immune responses to different parasite or host factors. […] Given the complexity and personalized immune response, we expect that much larger sample size, or a population meta-analysis, would be needed to identify any pattern of humoral response associated with host and parasite factors.”

In addition, we added more specifications on the volunteers in the Materials and methods, including the inclusion criteria to select malaria-negative patients, and the following paragraph:

“The 36 volunteers were healthy, adult males between 18-35 years old, with no parasitemia at the start of the study(measured by TBS and antibodies to PfEXP1 by ELISA), no history of malaria episodes over the last 5 years, and no parasitemia before CHMI (measured by TBS and qPCR) (Jongo et al., 2018). They were all students in Dar Es Salaam at the time of the study, however home town.”

The interesting impact from this research, I believe, is this assessment of possible pre-vaccination imprinting and what can we learn from this to improve vaccine strategies, including PfSPZ vaccine. I believe more should be focused on this given the results presented really had no clear post vaccination signature seen and protected vs. unprotected numbers were extremely small for wider conclusions on possible markers of protection. Exploring how different individuals with perhaps very similar exposure backgrounds come into a vaccine trial and if a vaccine can alter this pre-existing baseline to promote benefit is crucial to vaccine success and worth focusing more clearly on.

We thank reviewer 3 for this suggestion. We agree that no clear signature of protection was found, and we have reduced the paragraph discussing the potential association of the four proteins with protection highlighting the uncertainty around those findings (Discussion). We also highlighted that any signature of protection (single antigen reactivity and breadth) is based on a very small sample size and cannot be considered as a wider conclusion on possible markers or correlates of protection.

The Discussion now focuses primarily on the natural immune imprinting in pre-exposed individuals, and the potential impact on vaccine-induced protection. The third and fourth paragraphs would be the main paragraphs in the Discussion on natural imprinting. We have done additional literature research to contextualise in a broader view on the natural imprinting across different pathogens.

Additionally, our concluding paragraph hopefully better articulates the implications of our findings:

“Further reduction and eventual elimination of malaria requires significant investment and research and development of new tools, including vaccines or other immune therapies (Greenwood, 2008). […]Without further fundamental studies, additional hurdles for future vaccine trials remain in regards to the validity of extrapolating vaccine outcomes from trials in naïve cohorts to pre-exposed populations and different age groups.”